# Elucidating mechanisms of genetic cross-disease associations at the *PROCR* vascular disease locus

David Stacey[1,68], Lingyan Chen [1,68], Paulina J. Stanczyk[2,3], Joanna M. M. Howson [1,4], Amy M. Mason [1], Stephen Burgess [1,5], Stephen MacDonald[6], Jonathan Langdown[6], Harriett McKinney[7,8], Kate Downes[7,8,9], Neda Farahi[10], James E. Peters [1,11,12], Saonli Basu[13], James S. Pankow [14], Weihong Tang[14], Nathan Pankratz [15], Maria Sabater-Lleal [16,17], Paul S. de Vries[18], Nicholas L. Smith[19,20,21], CHARGE Hemostasis Working Group*, Amy D. Gelinas[22], Daniel J. Schneider [22], Nebojsa Janjic[22], Nilesh J. Samani [2,3], Shu Ye [2,3], Charlotte Summers [10], Edwin R. Chilvers [23], John Danesh[1,24,25,26,27] & Dirk S. Paul [1,24,27✉]

Many individual genetic risk loci have been associated with multiple common human diseases. However, the molecular basis of this pleiotropy often remains unclear. We present an integrative approach to reveal the molecular mechanism underlying the *PROCR* locus, associated with lower coronary artery disease (CAD) risk but higher venous thromboembolism (VTE) risk. We identify *PROCR*-p.Ser219Gly as the likely causal variant at the locus and protein C as a causal factor. Using genetic analyses, human recall-by-genotype and in vitro experimentation, we demonstrate that *PROCR*-219Gly increases plasma levels of (activated) protein C through endothelial protein C receptor (EPCR) ectodomain shedding in endothelial cells, attenuating leukocyte–endothelial cell adhesion and vascular inflammation. We also associate *PROCR*-219Gly with an increased pro-thrombotic state via coagulation factor VII, a ligand of EPCR. Our study, which links *PROCR*-219Gly to CAD through anti-inflammatory mechanisms and to VTE through pro-thrombotic mechanisms, provides a framework to reveal the mechanisms underlying similar cross-phenotype associations.

A full list of author affiliations appears at the end of the paper.

Genome-wide association studies (GWAS) have revealed widespread pleiotropy of disease-associated genetic variants. A recent study of cross-phenotype genetic association data in the UK Biobank has shown that 96% of trait-associated variants (minor allele frequency (MAF) ≥ 1%) are associated with more than one ICD-10 code, with some showing associations with more than 50 codes[1]. The vast majority of these pleiotropic variants were found to impact the risk of multiple diseases in a directionally consistent manner, but 1.9% of loci (excluding the major histocompatibility complex) showed evidence of both higher and lower risk effects attributable to the same allele[1]. One such example is rs9349379 A > G, a well-characterized regulatory variant at the *PHACTR1-EDN1* locus, which is associated with a higher risk of coronary artery disease but a lower risk of four other vascular diseases including migraine headache and hypertension[2].

Another example of a pleiotropic variant is p.Ser219Gly (rs867186 A > G) in the *PROCR* gene, which encodes the endothelial protein C receptor (EPCR), a key regulator of the protein C (PC) pathway. The minor G allele of this variant has been shown to correlate with a lower risk of CAD[3,4] and myocardial infarction[5], but a higher risk of venous thromboembolism (VTE)[6–8]. This pattern of opposing associations seems paradoxical because several conventional cardiovascular risk factors (e.g., measures of adiposity) show directionally concordant associations for CAD and VTE[9]. Further, GWAS of cardiovascular intermediate traits have reported associations between rs867186-G and components of the coagulation cascade, including higher plasma levels of PC[10] and coagulation factor VII[11,12]. However, the causal relevance of these intermediate traits to cardiovascular diseases remains uncertain.

The thrombomodulin–protein C pathway serves as a key mediator of the cross-talk between coagulation and inflammatory processes. It comprises molecular components that can respond to a range of pathophysiological environments in different vascular beds[13–15]. At the vascular endothelium, thrombomodulin binds to thrombin, directly inhibiting its clotting and cell activation potential and converting PC to activated PC (APC) (reviewed in[15,16]). The activation of PC by the thrombin–thrombomodulin (TM) complex is markedly enhanced when PC is presented by EPCR[17], a type I transmembrane protein that is mainly expressed on the endothelium of large blood vessels.[18,19] Once APC dissociates from EPCR, it binds to protein S to inactivate the coagulation factors Va and VIIIa, thereby inhibiting further thrombin generation. In addition, APC promotes fibrinolysis by decreasing the levels of plasminogen activator inhibitor type 1 (PAI-1), and reduces inflammation by inhibiting the production of tumor necrosis factor (TNF)-α and interleukin(IL)-1β (reviewed in[15,16]).

A soluble form of EPCR (sEPCR) is present in plasma, which is generated by ectodomain shedding of EPCR from the endothelium. Plasma sEPCR levels in healthy individuals display a bimodal distribution, with higher levels being associated with one of the four frequent haplotypes at the *PROCR* locus[8,20–23]. This haplotype (denoted A3 or H3) is tagged by the minor allele of the p.S219G variant. Functional studies showed that the variant results in increased shedding of EPCR from the endothelial surface by rendering the receptor more sensitive to cleavage by metalloprotease[21] and by forming an alternatively spliced, truncated transcript[24]. The shedding is effectively regulated by TNF-α and IL-1β[25]. sEPCR retains its ability to bind both PC and APC but does not enhance PC activation[26,27]. However, the precise molecular mechanism underlying the *PROCR*-p.S219G functional variant and its influence on the cardiovascular intermediate phenotypes that may mediate the risk of CAD and VTE is incompletely understood.

In this study, we aim (1) to systematically assess the association of the *PROCR*-p.S219G variant with a range of cardiometabolic outcomes and relevant risk factors; (2) to evaluate causality of individual components of the protein C pathway on cardiovascular diseases; and (3) to help uncover the molecular and cellular chain-of-events that connect the *PROCR*-219Gly allele to a lower risk of CAD but a higher risk of VTE. The results of our integrative epidemiological and functional analyses (Fig. 1) reveal new insights underlying the *PROCR* association locus for arterial and venous diseases and have potential implications for the development of therapeutic strategies targeting components of the protein C pathway.

## Results

**Association of *PROCR*-p.S219G with cardiovascular diseases and risk factors.** To search for associations of *PROCR*-p.S219G with a broad range of human diseases, we conducted a phenome-wide association analysis across 1402 electronic health record-derived ICD-codes from the UK Biobank. The association of *PROCR*-p.S219G with each of these codes was tested using SAIGE[28], a generalized mixed model association test that accounts for case-control imbalance and sample relatedness, as implemented in PheWeb (Methods). The data implicated diseases of the circulatory system, e.g., phlebitis/thrombophlebitis (PheWAS code 451; $P = 4.2 \times 10^{-8}$) and coronary atherosclerosis (PheWAS code 411.4; $P = 2.9 \times 10^{-5}$) (Fig. 2a).

Next, we performed a more focused phenome scan of the circulatory system. For each trait, we retrieved the largest available genetic association dataset (Methods; Supplementary Data 1). We found that the minor (G) allele of rs867186 (219Gly) was consistently associated with a higher risk of VTE in the UK Biobank (odds ratio (OR) = 1.15 [95% confidence interval (CI) = 1.10, 1.20]; $P = 1.87 \times 10^{-9}$) and INVENT (OR = 1.15 [1.07, 1.24]; $P = 1.21 \times 10^{-4}$) studies (Fig. 2b). We also observed a higher risk of deep vein thrombosis (DVT; OR = 1.18 [1.12, 1.25]; $P = 2.63 \times 10^{-10}$) and pulmonary embolism (OR = 1.13 [1.05, 1.21]; $P = 5.52 \times 10^{-4}$) in UK Biobank, with both of these

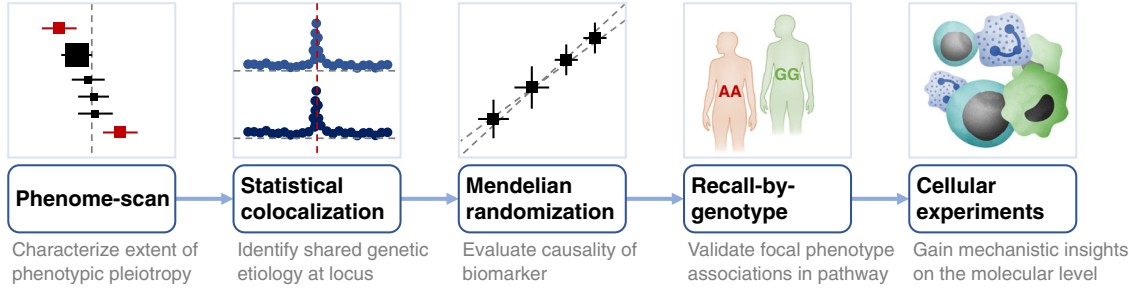

**Fig. 1 Schematic overview of the study design to elucidate molecular underpinnings of cross-disease associations.** Credits: The immune response, Big Picture (https://www.stem.org.uk/rx34vg).

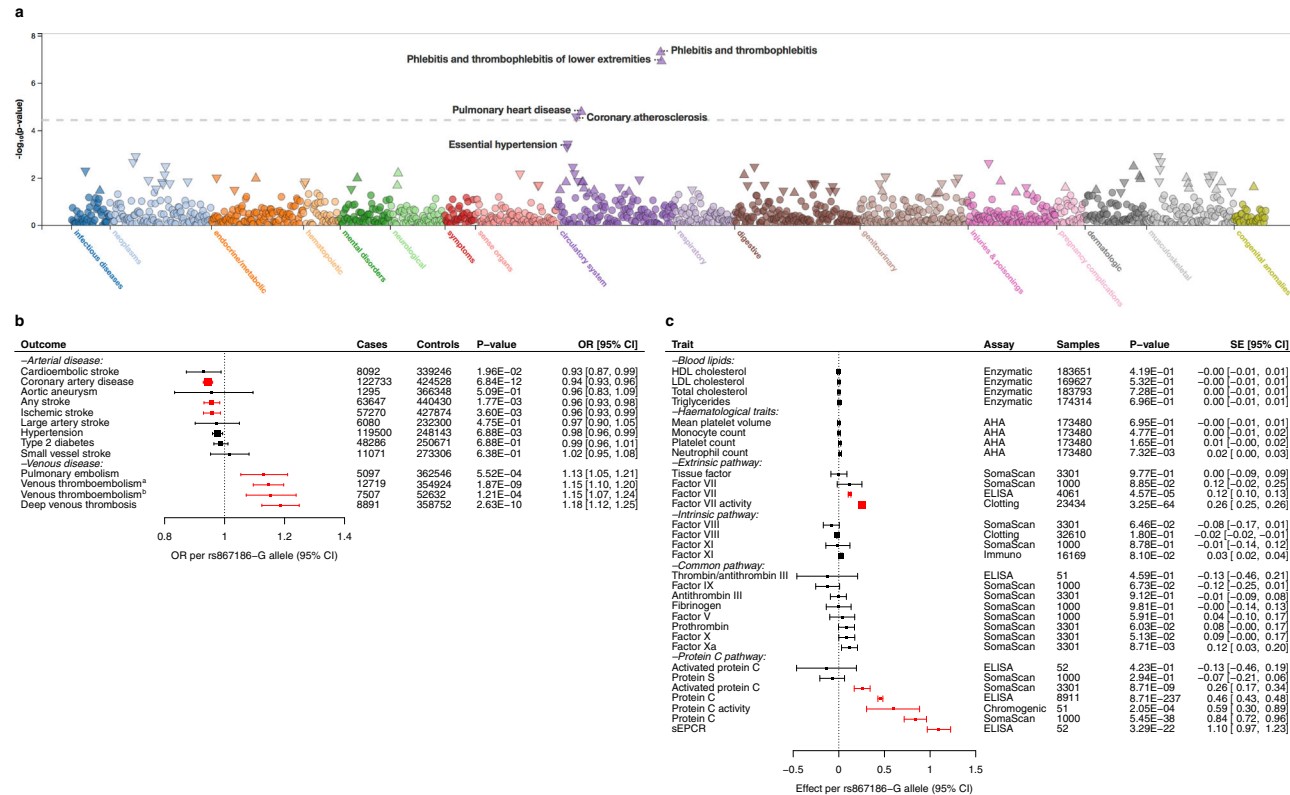

**Fig. 2 Association of *PROCR*-219Gly with a range of health outcomes and circulating cardiovascular biomarkers. a** Phenome-wide association scan of *PROCR*-p.S219G (rs867186) across 1402 broad electronic health record-derived ICD-codes from the UK Biobank. Unadjusted *P* values were obtained from the PheWeb portal. **b** Forest plot showing the associations of the minor (G) allele of rs867186 genotype with different cardiovascular conditions. Association statistics for VTE outcomes were obtained from the INVENT consortium (a) or UK Biobank (b). Data are presented as odds ratios with 95% confidence intervals (horizontal lines). Box sizes are proportional to inverse-variance weights. For each phenotypic subgroup, associations are ordered by their effect size. *P* values were obtained from the published GWAS. Associations that passed correction for multiple testing in this analysis (*P* = 0.05/13 traits = 3.85 × 10⁻³) are highlighted in red. The number of cases and controls for each association is shown in the forest plot. Supplementary Data 1 provides the association statistics for all traits, as well as data sources and references. **c** Forest plot showing the associations of rs867186-G with clinical biomarkers (blood lipids, hematological traits) and plasma proteins of the coagulation cascade (extrinsic, intrinsic and common pathways) and protein C pathways. Data are presented as per-allele changes in the traits expressed as standard deviations with 95% confidence intervals (horizontal lines). Box sizes are proportional to inverse-variance weights. For each phenotypic subgroup, associations are ordered by their effect size. *P* values were obtained from the published GWAS. Associations that passed correction for multiple testing in this analysis (*P* = 0.05/31 traits = 1.61 × 10⁻³) are highlighted in red. The number of participants for each association is shown in the forest plot. Supplementary Data 1 provides the association statistics for all traits, as well as data sources and references. Abbreviations: AHA automated hematology analyzer, ELISA enzyme-linked immunosorbent assay.

conditions being manifestations of VTE (Fig. 2b). In contrast, rs867186-G was associated with a lower risk of CAD in a large GWAS meta-analysis of the UK Biobank and CARDIoGRAM-plusC4D consortium (OR = 0.94 [0.93, 0.96]; $P = 6.84 \times 10^{-12}$) (Fig. 2b). Further, we detected a tentative association of rs867186-G with a lower risk of 'any' stroke (OR = 0.96 [0.93, 0.98]; $P = 1.77 \times 10^{-3}$) and ischemic stroke (OR = 0.96 [0.93, 0.99]; $P = 3.60 \times 10^{-3}$) in the MEGASTROKE consortium (Fig. 2b). Collectively, these data suggest that individuals carrying rs867186-G alleles have lower susceptibility to arterial thrombotic diseases but a higher risk of venous diseases.

To explore the molecular basis for this association pattern, we associated the rs867186-G allele with various intermediate traits related to the cardiovascular system (Methods). In particular, we focused on traits that directly influence the protein C pathway. We found that rs867186-G correlates strongly with higher PC levels in plasma, measured using an enzyme-linked immunosorbent assay (ELISA) in the ARIC study (per-allele effect = 0.46 standard deviation (SD) [0.43, 0.48]; $P = 8.71 \times 10^{-237}$) (Fig. 2c). The allelic effect was also observed

using the highly sensitive, multiplexed SomaScan assay in the KORA study (0.84 SD [0.72, 0.96]; $P = 5.45 \times 10^{-38}$). This assay quantifies the relative concentrations of plasma proteins or protein complexes using modified aptamers ('SOMAmer reagents')[29,30]. Further, the allele was significantly associated with elevated plasma levels of APC (0.26 SD [0.17, 0.34]; $P = 8.71 \times 10^{-9}$) and activity of coagulation factor VII (0.26 SD [0.25, 0.26]; $P = 3.25 \times 10^{-64}$) (Fig. 2c). We neither detected associations with plasma levels of other measured proteins in the coagulation cascade and protein C pathway, including protein S (the cofactor of APC), nor with risk factors for thrombosis, including fibrinogen, von Willebrand factor (vWF), plasminogen activator inhibitor-1 (PAI-1) and the thrombolytic agent tissue plasminogen activator (tPA) ($P > 0.05$)[31–34] (Fig. 2c). Finally, rs867186-G was not associated with conventional cardiovascular risk factors, including lipid levels, type 2 diabetes and hypertension (Fig. 2b, c).

We investigated a subset of the molecular intermediate traits, including PC, APC and FVII, using the SomaScan assay. To confirm the specificity of the binding events, we measured the

binding activity of the PC and APC SOMAmer reagents to a range of relevant proteins, specifically, PC, APC, sEPCR, thrombin, FV, FVIIa, protein S and thrombomodulin (Methods). We confirmed that the APC SOMAmers bind the proteins in a specific manner. However, we found that the PC SOMAmer binds to both the zymogenic and activated form of protein C (Supplementary Table 1), which may contribute to the observed difference in the magnitude of effect sizes observed for the immuno- and SomaScan assays (Fig. 2c). Additionally, we confirmed that the presence of relevant binding partners of PC and APC do not interfere with SOMAmer binding (Supplementary Table 1).

### Identification of shared genetic etiology at the *PROCR* locus.
Despite data showing associations of the rs867186 variant at the *PROCR* locus with CAD and VTE, it has been uncertain whether they reflect a shared causal variant and mechanism. To address this, we performed statistical colocalization analyses. We applied a Bayesian algorithm, Hypothesis Prioritization in multi-trait Colocalization (HyPrColoc)[35], which allows for the assessment of colocalization across multiple complex traits simultaneously (Methods). We found colocalization of the genetic association data of CAD and DVT as well as factor VII, PC and APC levels at the *PROCR* locus, with a posterior probability of colocalization of 99.37% (Fig. 3a). The variant rs867186 was found to be the likely causal variant at the locus explaining 99.31% of the posterior probability (Fig. 3b). Thus, these data provide support for a common genetic mechanism underlying the *PROCR* locus.

### Causal evaluation of protein C in arterial and venous diseases.
The association data suggest that genetic variants at the *PROCR* locus influence PC and APC abundance, FVII activity and susceptibility to CAD and DVT. However, these data do not necessarily imply that these molecular traits have a causal relationship with the disease phenotypes. To help define this relationship, we conducted Mendelian randomization (MR) analyses, using genetic variants as instrumental variables to avoid confounding and reverse causation[36]. We constructed a multi-allelic genetic score to estimate the causal associations between the putative risk factors and cardiovascular outcomes (Methods). The score comprised of approximately independent ($r^2 < 0.1$) SNPs at the *PROCR* region with $P$ value $\leq 5 \times 10^{-8}$ (Methods; Supplementary Table 2). Our data showed that every genetically-predicted increment (per 1 SD) in PC levels is associated with a lower risk of CAD (OR = 0.88 [0.86, 0.90]; $P = 4.17 \times 10^{-24}$), 'any' stroke, ischemic stroke and cardioembolic stroke, as well as a higher risk of VTE (OR = 1.24 [1.17, 1.32]; $P = 1.05 \times 10^{-11}$), DVT (OR = 1.34 [1.25, 1.44]; $P = 8.70 \times 10^{-16}$) and pulmonary embolism (Table 1; Supplementary Fig. 1). We also performed these analyses with APC, resulting in similar effect sizes and association $P$ values (Supplementary Table 3). Findings were robust to the use of a range of different MR approaches, i.e., inverse-variance weighting (IVW) method, median-based methods (simple and weighted) and MR-Egger regression (Methods). We conducted further sensitivity analyses, confirming the validity of our results to potential violations of the MR assumptions (Methods; Supplementary Fig. 2). We applied reverse MR to evaluate evidence for causal effects in the reverse direction by modeling disease phenotypes as the exposure and PC or APC level as the outcome using genome-wide significant predictors of disease (Methods). These analyses revealed no reverse causality of CAD or DVT/VTE on the levels of PC or APC (Table 1; Supplementary Table 3). Taken together, these analyses provide evidence of causal relationships between the levels of zymogenic

and activated protein C and CAD and VTE outcomes, in opposite directions.

### Validation of 'focal' phenotype associations in the protein C pathway.
To determine the molecular and cellular effects of rs867186, the causal variant at the *PROCR* locus, we performed a recall-by-genotype study. Such recall-studies allow for the strict control of experimental conditions (e.g., identical processing of blood samples), statistical efficiency (i.e., balanced recruitment based on genotype independent of MAF) and deep-phenotypic characterization of the collected samples (e.g., in vitro challenge experiments) (reviewed in[37]). From a genotyped panel of healthy volunteers, we selected 52 individuals stratified by rs867186 genotype and matched for sex and age (Methods). In these individuals, we measured four biomarkers in plasma representing focal phenotypes that describe the functional state of the protein C pathway, i.e., levels of protein C (inferred from a chromogenic assay measuring PC activation in response to an exogenous stimulus), APC, sEPCR and thrombin-antithrombin (TAT) complex (Methods). We found that the minor (G) allele of rs867186 associated with higher plasma levels of sEPCR ($\beta = 1.10$, $P = 3.29 \times 10^{-22}$) (Fig. 4a). This finding is consistent with previous reports[8,23,38–42]. We also found that the G allele associated with elevated PC activity, a marker for PC levels ($\beta = 0.59$, $P = 2.05 \times 10^{-4}$) (Fig. 4a). These data are concordant with and complementary to the data that we report from the epidemiological studies above (Fig. 2c), in that the chromogenic assay used here is not affected by potential binding-affinity effects of protein-altering variants often detected in protein-binding assays. We did not observe genotypic effects on plasma levels of either APC ($\beta = -0.14$, $P = 0.42$) or TAT complex ($\beta = -0.13$, $P = 0.46$) (Fig. 4a). Together, these data provide a direct comparison of the genotypic effect of the *PROCR* causal variant on the functional PC pathway, and independent confirmation that both sEPCR and PC are higher in carriers of *PROCR*-rs867186-G.

### Quantification of EPCR expression and shedding in endothelial cells.
We next aimed to identify the direct upstream molecular determinants of elevated sEPCR levels due to the rs867186-G genotype. Using transcriptomic data across 27 mature hematopoietic cell types from the BLUEPRINT Blood Atlas[43], we found that *PROCR* is highly expressed in human umbilical vein endothelial cells (HUVECs) and modestly expressed in macrophages, but not expressed in any other cell type analyzed (Supplementary Fig. 3). Consistent with these data, in flow cytometric analyses, we determined high expression of membrane-bound EPCR in HUVECs (Supplementary Figs. 4, 5). We detected 1.9-fold lower levels of EPCR in untreated HUVECs obtained from homozygotes of the rs867186-G-allele compared to homozygotes of the A-allele ($P = 0.0051$) (Fig. 4b). In HUVECs treated with phorbol 12-myristate 13-acetate (PMA), a potent agent to enhance ectodomain shedding, we found lower levels of EPCR compared to HUVECs treated with vehicle control in both homozygote groups, i.e., 4.1-fold ($P = 0.0124$) and 5.3-fold ($P = 0.0046$) for rs867186-G-allele and -A-allele homozygotes, respectively (Fig. 4c). Taken together, these findings are consistent with increased EPCR shedding from endothelial cells in carriers of *PROCR*-rs867186-G. We also performed flow cytometric analyses in a monocytic cell line (U937 cells), for which we observed modest levels of EPCR expression (Supplementary Fig. 6). We then differentiated these cells into macrophage-like cells using PMA and showed a ~30% reduction in EPCR expression relative to the undifferentiated cells (Supplementary Fig. 6). Finally, we sought to determine whether rs867186-G also affects EPCR shedding on primary neutrophils and monocytes purified from the individuals from our recall-study. However, we did

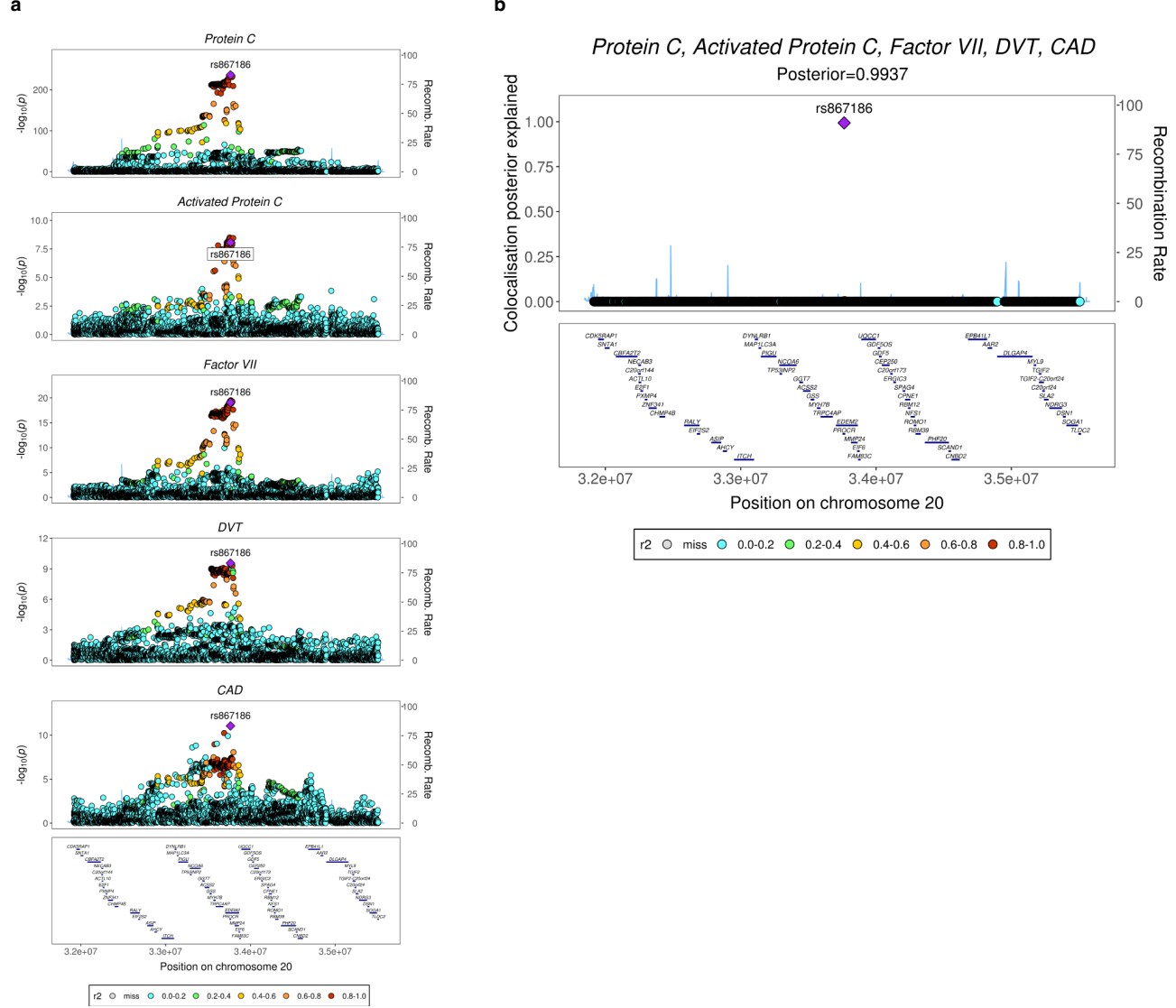

**Fig. 3 Statistical colocalization of cardiovascular outcomes and traits at the *PROCR* locus. a** Regional association plots at the *PROCR* gene locus, showing the genetic association with coagulation factor VII, protein C, activated protein C, DVT and CAD. Unadjusted *P* values were obtained from the published GWAS. Details about the statistical analysis and source of the data are given in the Methods section. Color key indicates r² with the respective lead variants in the GWAS. **b** Plot showing the colocalization posterior probabilities explained by each of the genetic variants at the chr20q11.22 locus tested in the colocalization analysis.

not detect the presence of EPCR on the surface of either of these cell types by flow cytometry (Supplementary Figs. 7, 8).

**Effect of sEPCR on leukocyte–endothelial cell adhesion**. Leukocyte–endothelial cell adhesion is a critical step in atherosclerosis that triggers vascular infiltration of monocytes and subsequently leads to microvascular inflammation[44]. Previous in vitro studies have highlighted EPCR as a potential modulator of the leukocyte–endothelial cell interaction. Specifically, sEPCR is a binding partner for the integrin macrophage-1 antigen (Mac-1)[45], which is expressed on the surface of activated leukocytes and is a key mediator of adhesion to the endothelium. Consequently, we investigated the effects of increasing concentrations of recombinant human sEPCR on leukocyte–endothelial cell adhesion using an in vitro static adhesion model. In brief, U937 cells were differentiated into macrophage-like cells using PMA and then dispensed onto a monolayer of TNF-α-activated HUVECs (Methods). Cell adhesion events were quantified following incubation with

increasing concentrations of anti-Mac-1 antibody (positive control) and recombinant sEPCR (Methods). We found that increasing concentrations of anti-Mac-1 antibody (compared to an IgG control; $P = 0.029$), but not sEPCR ($P > 0.05$) (Supplementary Fig. 9) led to a reduction of adhesion events.

**Effect of APC on cell adhesion molecule expression and leukocyte–endothelial cell adhesion**. Inflammatory cytokines such as TNF-α activate endothelial cells by increasing the expression of cellular adhesion molecules. We investigated whether human plasma-derived APC mitigates the TNF-α-associated increase in gene expression of cellular adhesion molecules, such as intercellular adhesion molecule 1 (ICAM-1) and vascular cell adhesion molecule 1 (VCAM-1). Using reverse transcription quantitative PCR (RT-qPCR), we showed that increasing concentrations of APC attenuate the TNF-α-induced increase in *ICAM1* mRNA levels in both HUVECs ($P = 0.0003$) and human coronary artery endothelial cells (HCAECs) ($P = 0.0009$) but not

**Table 1 Mendelian randomization estimates for the effect of genetically determined levels of protein C on the risk of vascular diseases and traits.**

| Exposure | Outcome | Number of SNPs[a] | MR causal estimate (IVW) | | Heterogeneity | |
|---|---|---|---|---|---|---|
| | | | Odds ratio [95% CI][b] | P value | Q-statistic | P value |
| Forward MR: | | | | | | |
| Protein C | Coronary artery disease | 19 | 0.88 [0.86, 0.90] | $4.17 \times 10^{-24}$ | 17.24 | 0.51 |
| Protein C | Deep venous thrombosis | 18 | 1.34 [1.25, 1.44] | $8.70 \times 10^{-16}$ | 14.21 | 0.65 |
| Protein C | Venous thromboembolism | 18 | 1.24 [1.17, 1.32] | $1.05 \times 10^{-11}$ | 17.80 | 0.40 |
| Protein C | Any stroke | 18 | 0.90 [0.86, 0.94] | $2.86 \times 10^{-6}$ | 13.79 | 0.68 |
| Protein C | Ischemic stroke | 18 | 0.90 [0.86, 0.95] | $3.77 \times 10^{-5}$ | 12.48 | 0.77 |
| Protein C | Pulmonary embolism | 18 | 1.17 [1.06, 1.29] | $2.65 \times 10^{-3}$ | 19.01 | 0.33 |
| Protein C | Cardioembolic stroke | 18 | 0.85 [0.77, 0.94] | $2.10 \times 10^{-3}$ | 17.83 | 0.40 |
| Protein C | Small-vessel stroke | 18 | 0.94 [0.82, 1.07] | 0.332 | 21.24 | 0.22 |
| Protein C | Large-artery stroke | 18 | 0.94 [0.83, 1.07] | 0.376 | 15.86 | 0.53 |
| Reverse MR: | | | | | | |
| Coronary artery disease | Protein C | 157 | 0.99 [0.95, 1.02] | 0.410 | 168.93 | 0.23 |
| Deep venous thrombosis | Protein C | 20 | 1.05 [0.91, 1.21] | 0.497 | 19.78 | 0.41 |
| Venous thromboembolism | Protein C | 21 | 1.16 [1.00, 1.34] | 0.050 | 15.04 | 0.77 |

[a]Number of SNPs as instrumental variants for PC.
[b]Represents increase/decrease of risk per SD increase in PC levels.
Effect estimates and P values are provided for the inverse-variance weighting (IVW) method. Q-statistic and respective P values are shown from the Cochran's Q-test for heterogeneity. Full details of the results from the different MR analyses, including details of data sources and number of cases, are reported in Supplementary Table 3.

*VCAM1* and *CCL2* mRNA levels ($P > 0.05$) (Fig. 5a). Notably, APC exposure also reduced *PROCR* gene expression in HUVECs ($P = 0.0066$) (Fig. 5a). Finally, in static leukocyte–endothelial cell adhesion assays, we showed that APC treatment leads to a reduction of leukocyte–endothelial cell adhesion events in HUVECs ($P = 0.0011$) and HCAECs ($P = 0.0246$) (Fig. 5b). Together, these data suggest that in carriers of the *PROCR*-219Gly genotype, who exhibit elevated APC levels (as measured on the SomaScan platform), the lower genetic susceptibility to arterial disease may be due to a reduced number of leukocyte–endothelial cell adhesion events at sites of vascular inflammation.

## Discussion

Elucidation of the molecular basis of cross-disease associations affords a major opportunity to advance understanding of disease etiology. Leveraging recent advances in population biobanks, statistical genomics and translational epidemiology, we illustrate an integrative, multi-modal approach to address this challenge. We applied this approach to two vascular diseases oppositely associated with the missense variant p.S219G (rs867186) in *PROCR*. We showed that *PROCR*-219Gly protects against CAD but increases susceptibility to VTE through distinct chains of molecular events, summarized in Fig. 6.

The data from our study show that *PROCR*-219Gly leads to a perturbed PC pathway, which acts focally to modulate the circulating levels of APC and has downstream effects on the biological mechanisms of associations with VTE and CAD.

We found that *PROCR*-219Gly is associated with higher circulating plasma sEPCR and lower EPCR levels on endothelial cells (Fig. 4), which is consistent with an increase in membrane shedding of EPCR and confirms findings from previous studies[8,21,38,42]. As only the membrane-bound form of EPCR is capable of activating PC[26], we anticipated that this reduction in EPCR would result in increased PC but reduced APC levels. Accordingly, we (Fig. 4a) and others[46,47] have observed higher plasma PC levels in *PROCR*-219Gly carriers. However, in our phenome-scan, we observed an unexpected increase in APC levels as measured on the SomaScan platform (Fig. 2c). We performed extensive testing to confirm the specificity of the APC SOMAmer (Supplementary Table 1), indicating that this finding is not due to cross-reactivity with PC or other coagulation factors.

Since the primary driver of PC activation in vivo is the TM complex[48], not EPCR, the higher levels of APC observed in *PROCR*-219Gly carriers may be due to an upregulation of TM activity in these individuals. In this scenario, an increase in APC would represent a homeostatic mechanism attempting to compensate for the increased thrombotic potential in *PROCR*-219Gly carriers and may be indicative of an acquired APC resistance. Indeed, APC resistance in the absence of Factor V Leiden is estimated to be prevalent in the general population (10–15%)[49]. Alternatively, given both PC and APC bind to sEPCR with the same affinity as EPCR[26], it is conceivable that the higher sEPCR levels observed in *PROCR*-219Gly carriers may serve to stabilize and prolong the presence of PC/APC in the circulation. This would be particularly salient for APC given its short half-life of ~15 minutes[50]. Furthermore, when bound to sEPCR, APC is unable to inactivate FV or FVIII[26,51]. Therefore, the sequestering of APC by sEPCR in *PROCR*-219Gly carriers may inhibit the anticoagulant activity of APC, resulting in APC resistance and increased thrombotic potential in these individuals. However, we did not observe statistically significant associations of *PROCR*-219Gly with FV or FVIII levels in our phenome scan (Fig. 2c).

In addition to its well-known role as an anticoagulant, APC has also been shown to function as a cytoprotective and anti-inflammatory agent via the protease-activated receptor 1 (PAR-1)[52]. Indeed, TNF-treated endothelial cells exposed to APC have reduced mRNA and surface protein levels of key intercellular adhesion molecules, such as intercellular adhesion molecule-1 (ICAM-1) (Fig. 5a), which regulate the adhesion of leukocytes to the endothelium[53–57]. By performing static adhesion assays, we provided evidence that increasing concentrations of APC reduce the adhesion of activated monocytes to endothelial cells (Fig. 5b). Based on these data, we propose that the APC/PAR-1 signaling pathway may be critical in protecting against CAD by reducing leukocyte–endothelium adhesion and vascular inflammation in the coronary arteries of *PROCR*-219Gly carriers (Fig. 6).

Our findings have implications for therapeutic strategies targeting the PC pathway for vascular diseases. Despite early positive clinical data that proposed the use of Drotrecogin alfa (Xigris®; a recombinant form of APC) as a therapeutic intervention for sepsis and septic shock, the medicine was withdrawn due to the lack of replication in subsequent trials and its associated risk of

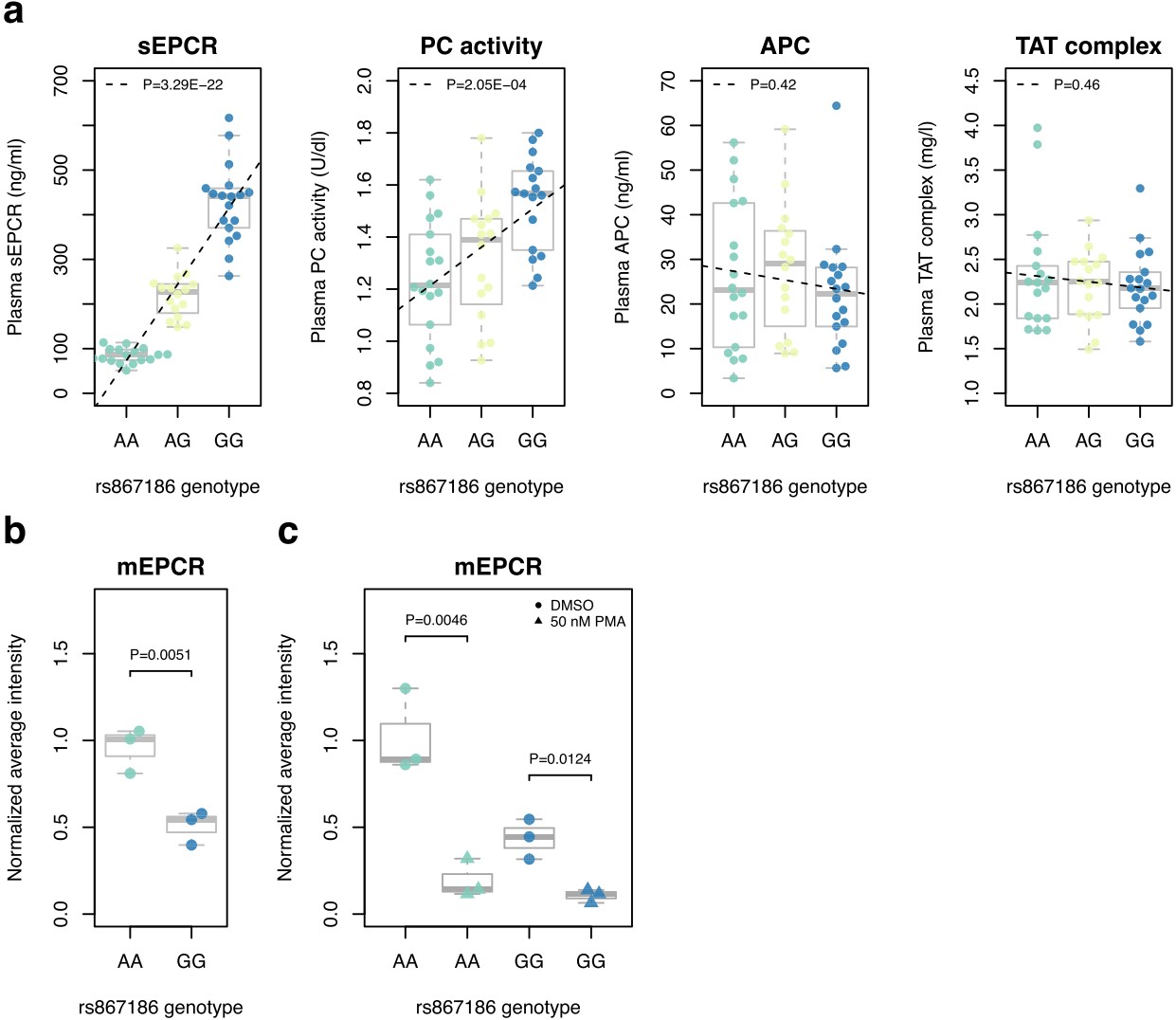

**Fig. 4 Effect of rs867186 genotype on plasma biomarkers and EPCR expression on HUVECs. a** Boxplots showing the distribution of plasma biomarker levels as a function of rs867186 genotype in up to 52 individuals. We measured plasma levels of sEPCR ($n = 52$ individuals across the three genotypic groups), APC ($n = 52$) and TAT complex ($n = 51$) using immunoassays, and PC levels ($n = 51$) using a chromogenic assay. All measurements were done with three technical replicates. The boxplots show the interquartile range in the box with the median as a horizontal line. Whiskers extend to ±1.5 times the interquartile range. Dashed lines indicate the fitted linear regression model for biomarker~genotype. *P* values for the additive regression model are indicated. **b** Boxplots showing the distribution of EPCR levels on HUVECs homozygous for the rs867186-G-allele or A-allele ($n = 3$ cell lines per genotypic group). Data show mean fluorescence intensity values of EPCR on untreated HUVECs, normalized to mean fluorescence intensity values of homozygotes of the rs867186-A-allele. The boxplot shows the interquartile range in the box with the median as a horizontal line. Whiskers extend to ±1.5 times the interquartile range. *P* values were calculated using a one-tailed *t*-test. **c** Boxplots showing the distribution of EPCR levels on HUVECs homozygous for the rs867186-G-allele or A-allele ($n = 3$ cell lines per genotypic group). Data show mean fluorescence intensity values of EPCR on HUVECs simulated with DMSO (vehicle control) or 50 nM phorbol 12-myristate 13-acetate (PMA), normalized to mean fluorescence intensity values of homozygotes of the rs867186-A-allele. The boxplots show the interquartile range in the box with the median as a horizontal line. Whiskers extend to ±1.5 times the interquartile range. *P* values were calculated using a paired one-tailed *t*-test. All experiments were performed with three technical replicates per cell line. Membrane EPCR levels were quantified using flow cytometric analysis (Methods). Bold lines and boxes represent the median and interquartile range of the data, respectively.

bleeding[58]. However, APC has since emerged as a potential candidate for the treatment of stroke. Clinical trials are currently ongoing to test in patients with acute ischemic stroke the efficacy of 3K3A-APC, a recombinant form of APC that lacks its anticoagulant activity but retains its PAR-1 cell-signaling activities[59]. Preliminary results showed that patients receiving 3K3A-APC had reduced hemorrhage volume and hemorrhage incidence on day 30 following initial drug infusion, relative to a placebo group[60]. The findings from these clinical studies are consistent with the results of our wide-angled genetic association scan

(Fig. 2), and provide a rationale to define and catalogue the disease relationships of pleiotropic variants on a genome-wide level to inform the development of new medicines.

The presented phenome-scan also showed a significant association of *PROCR*-219Gly with higher plasma levels of FVII (Fig. 2c). Recently, FVII has been identified as a ligand for EPCR and shown to bind EPCR with the same affinity as PC[61]. Although EPCR does not affect the activation of FVII, the interaction of EPCR with FVII leads to the clearance of FVII/FVIIa from the circulation through endocytosis[61]. Our data are

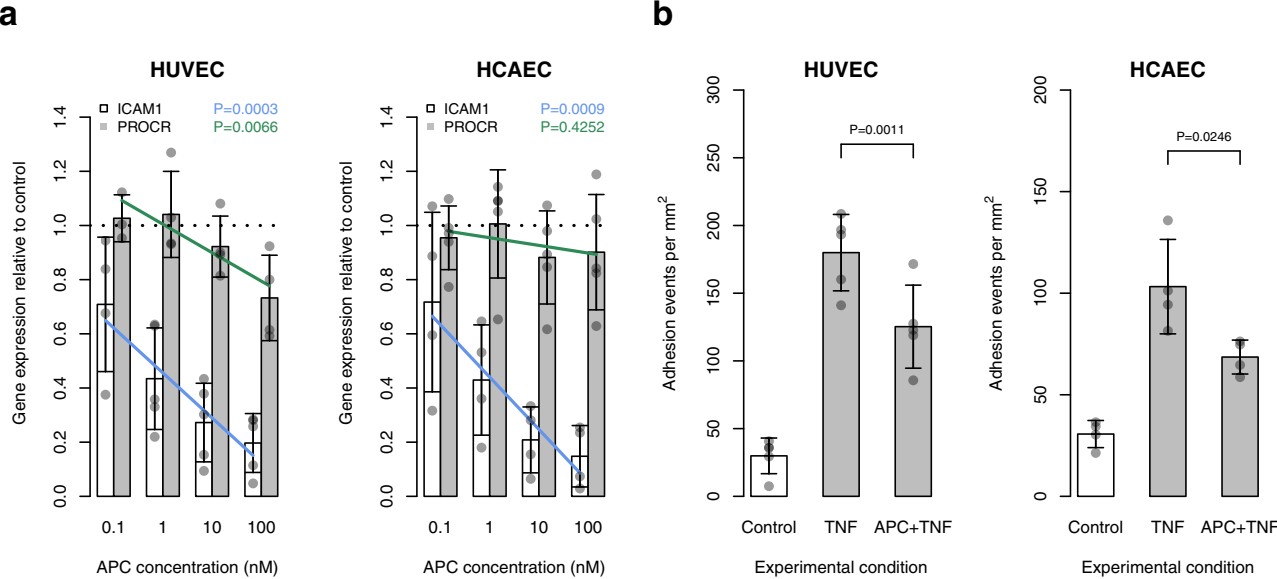

**Fig. 5 Effect of APC on cell adhesion molecule expression and leukocyte–endothelial cell adhesion. a** Barplots showing gene expression levels of *ICAM1* and *PROCR* in human umbilical vein endothelial cells (HUVECs) and human coronary artery endothelial cells (HCAECs) relative to the control condition (i.e., 0 nM APC; indicated with a dashed line). Cells were co-incubated with 1 ng/ml TNF-α and varying concentrations of APC (0, 0.1, 1, 10, 100 nM) for 24 h. Data are shown for $n = 5$ (*ICAM1*) and $n = 4$ (*PROCR*) biological replicates in HUVECs and $n = 4$ (*ICAM1*) and $n = 5$ (*PROCR*) biological replicates in HCAECs. Error bars show standard deviations of the means. The blue and green lines indicate the fitted linear regression model for gene expression level–log(APC concentration). *P* values for the F-test of the linear regression model are shown. For each biological replicate, three technical replicates were averaged. **b** Barplots showing mean cell adhesion events using static adhesion assays with PMA-stimulated monocytic cells (U937) and TNF-α-activated HUVECs or HCAECs. Cells were co-incubated with 1 ng/ml TNF-α and 100 nM APC for 24 h (Methods). Data are shown for $n = 5$ and $n = 4$ biological replicates in HUVECs and HCAECs, respectively. Error bars show standard deviations of the means. *P* values were calculated using paired one-tailed *t*-tests. For each biological replicate, 2–4 technical replicates were averaged.

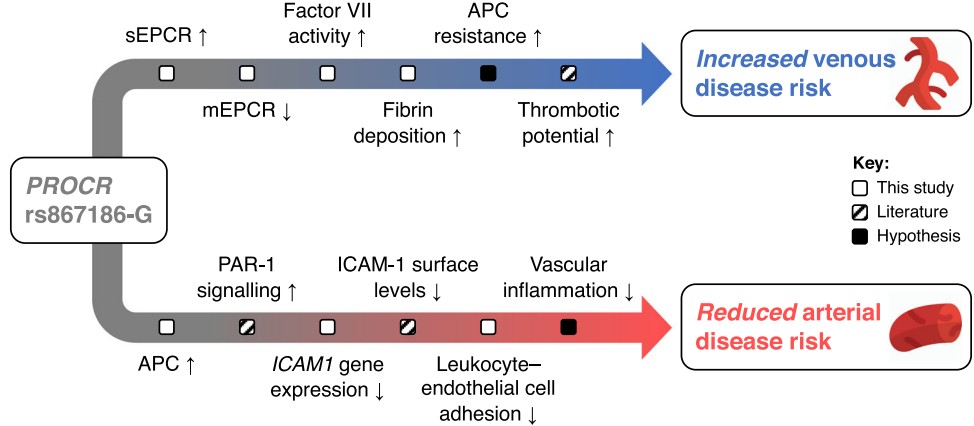

**Fig. 6 Proposed molecular mechanism underlying the *PROCR*-p.S219G variant.** Credits: Icons were made by Pixel perfect from https://www.flaticon.com/.

consistent with this observation, as *PROCR*-219Gly is not only associated with higher plasma levels of FVII but also reduced levels of EPCR (Fig. 4b, c). Thus, the reduced availability of EPCR could directly contribute to the reduced internalization of FVII/FVIIa and increased accumulation in the circulation, which in turn may increase thrombotic potential. Further research is necessary to confirm that *PROCR*-219Gly is indeed associated with reduced FVII/FVIIa internalization, for example, through performing endocytosis assays in genotype-specific or CRISPR/Cas9-edited endothelial cell lines. Nevertheless, this proposed mechanism is consistent with the suggestive genetic association signals observed in *PROCR*-219Gly carriers for higher levels of D-dimer[62], a marker of blood clot degradation, and shorter prothrombin time[63] ($P = 3.70 \times 10^{-6}$ and $P = 9.98 \times 10^{-8}$, respectively). The association with shorter prothrombin time was replicated at genome-wide significance in the Japanese population ($P = 5.64 \times 10^{-24}$)[64].

We acknowledge that our study has limitations. First, many hemostatic factors have short half-lives[50,65], which presents a technical challenge for studies seeking to quantify accurately these markers. Second, contrary to the statistically significant association between rs867186 and APC levels as measured using the SomaScan assay, in our recall-study, we found no evidence of an association. This is likely due to the difference in statistical power between the two experiments, with sample sizes of 3,301 and 52 individuals, respectively (Supplementary Fig. 10). Indeed, previous studies that aimed to ascertain an association between rs867186 and APC have been hampered by relatively small sample sizes, yielding mixed findings[8,39,40]. Replication in independent large cohorts is needed. Third, EPCR is expressed on the

surface of platelets[66]. As the blood processing in our study likely resulted in platelet-poor as opposed to platelet-free plasma, it is possible that some of the sEPCR signal observed in our recall experiment (Fig. 4a) may have originated from platelet-associated EPCR. However, *PROCR* mRNA levels are very low in human platelets (Supplementary Fig. 3), suggesting that any platelet-associated signal is likely to be negligible. Fourth, further studies are required to elucidate the complex interactions between EPCR and its ligands PC (APC) and FVII (FVIIa), as well as the downstream consequences of these interactions on hemostasis.

Several aspects of our approach are generalizable to the study of other cross-disease associations (Fig. 1). First, the availability of large, disease-agnostic population biobanks with linked genomic, molecular phenotype and health record data, such as UK Biobank, provides an opportunity to systematically characterize the molecular underpinnings of health outcomes. Second, publicly available bioinformatics tools, including SAIGE[28] and PhenoScanner[67], allow for the mining of these data and the generation of specific hypotheses about the underlying biological mechanisms at individual genetic association loci. Third, the use of freely available software for statistical colocalization[35] and Mendelian randomization[68] analyses enables evaluation of the extent to which associated phenotypes share the same causal variant and the causal relationship of molecular biomarkers with a disease outcome. Fourth, the increasing availability of volunteers in bioresources (e.g., UK National Institute for Health Research BioResource) who have agreed to participate in biomedical studies on the basis of their genetic and/or phenotypic characteristics enables targeted mechanistic studies tailored to specific hypotheses. This includes recall-by-genotype studies, which afford an efficient approach to detailed phenotyping that can be applied to different study designs, biological samples and experimental techniques[37].

Taken together, our study provides new insights into the role of the PC pathway in arterial and venous diseases. We demonstrate that the combination of population biobank data and advanced statistical methods can help identify causal biomarkers and pathways, and that recall-by-genotype is a powerful experimental approach that can yield informative mechanistic insights. Overall, our study provides a framework for mapping molecular mechanisms that underlie cross-phenotype associations.

## Methods

***PROCR*-rs867186 phenome-scan**. The phenome-wide association scan of *PROCR*-p.S219G (rs867186) across electronic health record-derived ICD-codes from the UK Biobank was conducted using PheWeb v1.1.17 (http://pheweb.sph.umich.edu/SAIGE-UKB/variant/20:33764554-A-G). To assess the effects of *PROCR*-rs867186 genotype on cardiovascular intermediate traits and outcomes, we collated data from the latest available genome-wide association studies using PhenoScanner v2, a database of human genotype–phenotype associations[67]. To allow comparative analyses, we considered data from individuals of European ancestry where possible. We focused our analyses on cardiometabolic traits and outcomes; thus, not all genome-wide significant associations are reported. The following association statistics were retrieved: stroke outcomes from the MEGASTROKE consortium[69]; venous thromboembolism outcomes from INVENT consortium[7] or UK Biobank; hypertension and aortic aneurysm from UK Biobank; coronary artery disease from van der Harst et al.[4]; type 2 diabetes from Mahajan et al.[70]; blood lipids from the Global Lipids Genetics consortium[71]; hematological traits from Astle et al.[72]; and plasma proteins of the coagulation cascade and protein C pathway from the ARIC study[10], CHARGE consortium[34,73,74], Sun et al.[30] and Suhre et al.[75]. To enable a comparison of the magnitude of the effect sizes, we conducted analyses with standardized units of measurement for each quantitative trait. Supplementary Data 1 provides further details of all data used in our analyses. The data availability section provides further information on the results of the complete data query. Supplementary Data 2 shows an overview of the associations of *PROCR*-rs867186 with protein levels measured by the SomaScan platform.

**Determination of equilibrium binding constants**. Equilibrium binding constants ($K_d$ values) of modified aptamers were determined by filter binding assay. $K_d$ values

of modified aptamers were measured in SB18T buffer (40 mM Hepes pH 7.5, 102 mM NaCl, 5 mM KCl, 5 mM MgCl$_2$, 0.01% Tween-20). Modified aptamers were 5′ end-labeled using T4 polynucleotide kinase (New England Biolabs) and γ-[$^{32}$P]ATP (Perkin-Elmer). Commercially available proteins to be used in the filter binding assay (protein C, APC, sEPCR, thrombin, factor V, factor VIIa, protein S and thrombomodulin) were biotinylated by covalent coupling of EZ-Link NHS-PEG4 -Biotin (Thermo Scientific) following the manufacturer's protocol. Briefly, proteins were combined with a 10-fold molar excess of EZ-Link NHS-PEG4 -Biotin in SB18T buffer and incubated at room temperature for 30 min. Free biotin was removed via YM-3 filtration (Millipore). Following biotinylation, protein concentrations were determined using a Micro BCA Protein Assay kit (Thermo Fisher). Radiolabeled aptamers (~20,000 CPM, 0.03 nM) were mixed with biotinylated proteins at concentrations ranging from $10^{-7}$ to $10^{-12}$ M and incubated at 37 °C for 40 min. Bound complexes were partitioned on MyOne streptavidin beads (Invitrogen) and captured on Durapore filter plates (EMD Millipore). The fraction of bound aptamer was quantified with a phosphorimager (Typhoon FLA 9500, GE) and data were analyzed in ImageQuant TL (GE). To determine binding affinity, data were fit using the equation: y = (max − min)(Protein)/($K_d$ + Protein) + min.

**Competition binding assays**. Competition binding assays were performed to test whether sEPCR and thrombomodulin interfere with the SOMAmer reagent 2961-1_2 binding to protein C or whether sEPCR, protein S and factor VIIa interfere with the binding of SOMAmer reagents 2961-1_2, 3758-63_3 and 3758-68_3 to APC. These experiments were performed by pre-incubating equal volumes of biotinylated protein C (80 nM) or biotinylated APC (48 nM or 80 nM) with competitor protein concentrations ranging from $10^{-5}$ to $10^{-10}$ M at 37 °C for 30 min in SB18T buffer in the presence of 2 µM polyanionic competitor Z-block (a 30-mer modified DNA sequence, [AC(BndU)$_2$]$_7$AC)[23] to allow protein complexes to form. Following the 30-min incubation, the reaction was diluted in half with radiolabeled SOMAmer reagent (20,000–60,000 CPM, 0.03 nM) and returned to 37 °C for an additional 30 min. Bound complexes were partitioned on MyOne streptavidin beads and captured on Durapore filter plates. The amount of bound aptamer was quantified with a phosphorimager and data were analyzed in ImageQuant. The fraction of SOMAmer bound at each competitor concentration was normalized to the signal in the no competitor control well.

**Multi-trait colocalization**. We performed colocalization analysis at the *PROCR* gene locus (chr20: 31,916,110–35,505,723 bp; hg19), as defined based on recombination rates[3]. Details about the GWAS summary statistics used for this analysis are provided in Supplementary Data 1. Variants with both imputation (INFO)-score <0.7 and MAF < 0.01, or variants with INFO-score <0.3 and MAF > 0.01 were removed. The remaining 4,264 SNPs shared across each of the datasets were aligned to the DNA plus-strand (hg19) prior to colocalization analyses. We used a Bayesian algorithm, implemented in the Hypothesis Prioritization in multi-trait Colocalization (HyPrColoc) v1.0 method[35], to perform colocalization across all traits simultaneously. HyPrColoc extends the established colocalization methodology[76] by approximating the true posterior probability of colocalization with the posterior probability of colocalization at a single causal variant and a small number of related hypotheses[35]. If all traits do not share a causal variant, HyPr-Coloc employs a novel branch-and-bound selection algorithm to identify subsets of traits that colocalize at distinct causal variants at the locus. We used uniform priors as primary analysis and set strong bounds for the regional and alignment probabilities as default, i.e., the $P_R^*$ (regional probability threshold) = $P_A^*$ (alignment probability threshold) = 0.8, so that the algorithm identified a cluster of traits only if the posterior probability of full colocalization (PPFC) = $P_R P_A$ > 0.64. We also performed sensitivity analysis with non-uniform priors to assess the choice of priors, which used a conservative variant-level prior structure with $P = 1 \times 10^{-4}$ (prior probability of a SNP being associated with one trait) and $P_c = 1 - \gamma = 0.02$ ($P_c$ is the conditional colocalization prior that a SNP is causal for an additional trait given that it is causal for one trait), i.e., 1 in 500,000 variants is expected to be causal for two traits.

**Selection of instrumental variables for MR analysis**. We obtained regional association statistics at the *PROCR* region for plasma PC levels from the ARIC study and plasma APC levels from the INTERVAL study to assess the causal effects of PC (APC) on cardiovascular outcomes. Details about the GWAS data on cardiovascular outcomes are provided in Supplementary Data 1. To select genetic variants as instrumental variables for PC levels, we first removed SNPs with MAF < 0.01 and INFO-score <0.8. Next, we performed LD clumping to obtain approximately independent SNPs. In brief, the algorithm groups SNPs in LD ($r^2 \geq 0.1$ in 4,994 participants from the INTERVAL study[77]) within ±1 MB of an index SNP (i.e., SNPs with association $P$ value ≤5 × $10^{-8}$). The algorithm tests all index SNPs, beginning with the smallest $P$ value and only allowing each SNP to appear in one clump. Thus, the final output contains the most significant protein-associated SNPs for each LD-based clump across the genomic region. An overview of the instrumental variables is provided in Supplementary Table 2. This analysis was performed using PLINK v1.90[78].

**Mendelian Randomization analyses.** We used two-sample Mendelian randomization (MR)[68,79] to estimate the causal associations between PC and cardiovascular outcomes. The MR approach was based on the following assumptions: (i) the genetic variants used as instrumental variables are associated with PC levels; (ii) the genetic variants are not associated with any confounders of the exposure-outcome relationship; and (iii) the genetic variants are associated with the outcome only through changes in PC levels, i.e., a lack of horizontal pleiotropy. We applied the inverse-variance weighting (IVW) method in a multiplicative random-effect meta-analysis framework[79], MR median-based method (simple and weighted)[80], MR-Egger regression[81] and MR-PRESSO[82] to estimate the causal effects. We also performed several sensitivity analyses to assess the robustness of our results to potential violations of the MR assumptions, given these analyses have different assumptions for validity: (i) heterogeneity was estimated using the MR-IVW$Q$-statistic; (ii) horizontal pleiotropy was estimated using MR-Egger's intercept; (iii) the median-based methods have greater robustness to individual genetic variants with strongly outlying causal estimates compared with the inverse-variance weighted and MR-Egger methods; and (iv) influential outlier instrumental variables due to pleiotropy were identified using MR-PRESSO and (v) MR-Steiger filtering[83] was used to eliminate spurious results due to reverse causation. We also applied reverse MR[84] to evaluate evidence for causal effects in the reverse direction by modeling disease phenotypes as the exposure and PC or APC levels as the outcome. Instrumental variants for phenotypes of interest (i.e., CAD, DVT/VTE) were selected from their original GWAS data (Supplementary Data 1). The effects of these GWAS SNPs on PC levels were derived from Sun et al.[30]. The power and strength of the instrumental variables was assessed using the variance explained ($R^2$) and F-statistics (F = $\beta^2/se^2$)[85]. The MR analyses were conducted using the MendelianRandomization v0.3.0[68], TwoSampleMR v0.3.4[86] and MR-PRESSO v1.0[82] packages in R v3.4.2.

**Recall-by-genotype study.** The study was approved by the Leicester Central Research Ethics Committee and Health Research Authority (Reference: 17/EM/0028). Healthy volunteers were recruited from the NIHR Cambridge BioResource with informed consent. Participants who were older than 18 years of age and of European ancestry were selected based on $PROCR$-rs867186 genotype and homozygosity of the major allele for both $F5$-R506Q (rs6025; Factor V Leiden) and $F2$-G20210A (rs1799963; Factor II). Participants across the three rs867186 genotype groups were matched at the end of the study with respect to sex and age (within 10 years). Study participants were excluded that had a diagnosis of (i) a chronic disease; (ii) hypertension (or history of consistently high blood pressure readings, i.e., >140/90 mmHg); and/or (iii) hypercholesterolemia (or history of consistently high cholesterol levels, i.e., >6 mmol/l). Participants agreed to fast and abstain from caffeinated drinks for at least four hours prior to the study visit and to not receive any vasoactive medication for up to seven days prior to procedures. The study design was informed by a power calculation (Supplementary Fig. 10).

**Assessment of baseline characteristics of recall-by-genotype study participants.** Participants reported past medical conditions, demographic factors (e.g., ethnicity) and lifestyle factors (e.g., smoking and alcohol consumption). Height and weight/body fat were measured using a stadiometer and bioelectrical impedance (i.e., Tanita scale), respectively. Blood pressure and heart rate were assessed in one-min intervals using a validated, automated device while seated and again after 3–5 min standing. All measurements were done in triplicate using the same arm. An overview of the characteristics of the study participants is provided in Supplementary Table 4. These characteristics are presented as mean and standard deviation or percentage. Continuous and categorical variables between homozygous groups were compared using the two-sample $t$-test and chi-square test, respectively.

**Blood sample collection and processing.** A total of 46 ml of peripheral blood was collected from each donor in our recall-study using a 21 gauge needle unless clinically contraindicated. We collected blood in two S-Monovette 7.5-ml K3 EDTA tubes and two S-Monovette 10-ml sodium citrate 3.2% (1:10) 9NC tubes (Sarstedt). Samples were immediately centrifuged at 4 °C and 1000 × g for 15 min. Multiple aliquots of the top phases were stored at −80 °C within 30 min of blood draw. A full blood count for all donors was obtained from blood collected in a S-Monovette 1.2-ml K3 EDTA tube using a Sysmex Hematological analyzer.

**Quantification of plasma biomarkers.** Samples were thawed at 37 °C for 15 min, mixed and then centrifuged at room temperature and 3000 × g for 10 min immediately prior to assay. Soluble EPCR levels were determined using an Asserachrom sEPCR kit (00264; Diagnostica Stago); thrombin/antithrombin III complex levels using an Enzygnost TAT micro immunoassay (OWMG15; Siemens Healthcare Diagnostics Limited); APC levels using an Activated Protein C assay kit (CSB-E09909H; Cusabio Biotech); and PC levels using a HemosIL Protein C chromogenic assay (0020300500; Instrumentation Laboratory). All assays were performed according to the manufacturer's instructions. Samples were analyzed in random order and laboratory staff were blinded to genotype status. Participants with biomarker levels (or activity levels) 3 standard deviations above or below the population mean were removed.

**Tissue culture.** All cultures were maintained at 37 °C in a humidified chamber at 5% $CO_2$. For adhesion assays, Human Umbilical Vein Endothelial Cells (HUVECs) (PromoCell) and Human Coronary Artery Endothelial Cells (HCAECs) (Lonza) were cultured using an Endothelial Cell Growth Media (EGM)-Plus BulletKit (Lonza) and Microvascular Endothelial Cell Growth Medium-2 (EGM-2 MV) BulletKit (Lonza), respectively. U937 cells (ATCC) were suspended in RPMI-1640 Medium with GlutaMAX supplement, 10% fetal bovine serum (FBS), 100 U/ml penicillin and 100 U/ml streptomycin (ThermoFisher). HUVECs were used in experiments at passages 2–4, and U937 cells were discarded after passage 10. Throughout our study, we ensured cell viability of >95% using Trypan Blue. Genotype-specific HUVECs were prepared from tissues provided by the Anthony Nolan Trust Biobank. Three independent lines per genotype (i.e., rs867186-AA and -GG) were used for all experiments. Genotype-specific HUVECs were cultured in M199 Medium (Sigma) supplemented with 15% FBS (Sigma), 5 ng/ml Fibroblast Growth Factor-Acidic human (Sigma), 4.5 µg/ml Endothelial Cell Growth Supplement (Fisher Scientific), 10 U/ml heparin (Sigma) and 2.5 µg/ml thymidine (Sigma). Cells at passages 3–4 were used in experiments.

**Antibodies and recombinant human proteins.** Allophycocyanin (APC)-conjugated Rat Anti-Human EPCR monoclonal antibodies derived from two different clones were obtained from BD Biosciences (563622; clone: RCR-252) and Thermo Fisher Scientific (17-2018-42; clone: RCR-227). Corresponding APC-conjugated Rat IgG1, κ Isotype Control antibodies were sourced from BD Biosciences (554686; clone: R3-34) and Thermo Fisher Scientific (17-4301-82; clone: eBRG1). An unconjugated Rat Anti-Human EPCR monoclonal antibody was obtained from BD Biosciences (552500; clone: RCR-252), and an unconjugated Human CD11b/Integrin alpha M Antibody (anti-Mac1) (MAB1699; clone: ICRF44) was obtained from Bio-Techne. Fluorescein isothiocyanate (FITC)-conjugated mouse anti-human CD14 (325603; clone: HCD14) and CD16 (360715; clone: B73.1) antibodies were obtained from BioLegend. Recombinant human TNF-α (210-TA) and EPCR (9557-ER-050) were obtained from Bio-Techne, and plasma-derived APC (P2200) was obtained from Sigma-Aldrich.

**Quantification of EPCR levels on genotyped HUVECs.** Cultured HUVECs at baseline or treated with Phorbol myristate acetate (PMA, Sigma) or control DMSO (Sigma) were collected by trypsinization and then re-suspended in 1% BSA/PBS to a final concentration of 1 × $10^5$ cells/500 µl. Rat Anti-Human EPCR monoclonal antibodies and isotype controls (BD Biosciences) were added as appropriate at a final concentration of 0.125 µg/500 µl and incubated at room temperature for 20 min in the dark. Cells were washed once with cold 1% BSA/PBS and re-suspended in 1 ml ice-cold PBS prior to flow cytometric analysis using Gallios Flow Cytometer (Beckman Coulter) with Cytomics CXP software v2.2. Results were recorded as median fluorescence intensity and raw data were analyzed using Kaluza Analysis v1.3 (Beckman Coulter). We used one-tailed $t$-tests to test for differences in mean fluorescence intensities between the specific groups. We applied paired tests when comparing PMA vs vehicle control and unpaired when testing between genotypes.

**Quantification of EPCR levels on human monocytes and neutrophils.** We lysed 100-µl citrated whole blood samples at room temperature for 10 min using Lysing Solution 10X Concentrate (349202; BD Biosciences). Lysed blood was then centrifuged at 4 °C and 600 × g for 6 min, and the pellet re-suspended in HEPES buffered saline (Sigma-Aldrich). Cultured cells were also re-suspended in HEPES buffered saline, to a final concentration of $10^5$ cells/100 µl. Rat Anti-Human EPCR monoclonal antibodies and isotype controls were added as appropriate at a final concentration of 0.125 µg/100 µl and incubated at room temperature for 20 min in the dark. Samples were diluted in 0.5 ml ice-cold HEPES buffered saline prior to flow cytometric analysis using either a Cytomics FC500 with Cytomics CXP software v2.2 or a CytoFLEX S Flow Cytometer with CytExpert Acquisition and Analysis software v2.3 (Beckman Coulter). CD14+ Monocytes and CD16+ neutrophils from blood lysates were gated using forward and side light scatter, enabling discrimination by cell size and granularity, respectively. The gating strategy was validated using mouse anti-human CD14 and CD16 antibodies (Supplementary Fig. 7) added to blood lysates at a final concentration of 0.125 µg/100 µl. Results were recorded as median fluorescence intensity.

**Reverse transcription quantitative PCR (RT-qPCR).** HUVECs were seeded at a density of 31,250 cells/cm$^2$ (3 × $10^5$ cells per well of a 6-well plate) in 2 ml medium and left to attach overnight. Cells were then co-incubated with 1 ng/ml TNF-α and varying concentrations of APC (0, 0.1, 1, 10, 100 nM) for a further 24 hr prior to cell lysis and RNA extraction using a Quick-RNA Microprep kit (Zymo Research). RNA was quantified using a NanoDrop Lite Spectrophotometer (Thermo Scientific). 1 µg of RNA was reverse transcribed using a Maxima H Minus First-Strand cDNA Synthesis kit with dsDNase (Thermo Scientific), and cDNA was diluted 1:20 in ddH$_2$O. Quantitative PCR (qPCR) reactions were performed in solution containing 10 µl SYBR Green PCR Master Mix (Thermo Scientific), 70 nM of each forward and reverse primer, 4 µl cDNA and ddH$_2$O to a total volume of 20 µl. The sequences of all primers used in this study are as follows: $ACTB$: forward 5′-CCC TGG AGA AGA GCT ACG AG-3′, reverse 5′-GGA TGC CAC AGG ACT CCA T-

3′; *GAPDH*: forward 5′-CCC ACT CCT CCA CCT TTG AC-3′, reverse 5′-CCA CCA CCC CGT TGC TGT A-3′; *RPLP0*: forward 5′-GCA TCT ACA ACC CTG AAG TGC-3′, reverse 5′-TTG GGT AGC CAA TCT GCA GA-3′; *GUSB*: forward 5′-ACG TGG TTG GAG AGC TCA TT-3′, reverse 5′-TCT GCC GAG TGA AGA TCC C-3′; *ICAM1*: forward 5′-TGA TGG GCA GTC AAC AGC TA-3′, reverse 5′-GCG TAG GGT AAG GTT CTT GC-3′; *VCAM1*: forward 5′-TGT GAA GGA ATT AAC AG GCT G-3′, reverse 5′-TGA CAC TCT CAG AAG GAA AAG C-3′; *CCL2*: forward 5′-CAT GAA AGT CTC TGC CGC C-3′, reverse 5′-GGT GAC TGG GGC ATT GAT TG-3′; *PROCR*: forward 5′-CGG TAT GAA CTG CGG GAA TT-3′, reverse 5′-GTG TAG GAG CGG CTT GTT TG-3′. qPCR reactions were run using a QuantStudio 6 Flex Real-Time PCR instrument with QuantStudio software v1.3 (Thermo Scientific). After an initial step of 15 min at 95 °C, samples were subjected to 40 cycles of 30 sec at 95 °C and 30 sec at 59 °C, followed by dissociation curve analysis. Target $C_t$-values were normalized using the arithmetic mean of four endogenous control genes (*ACTB*, *GAPDH*, *RPLP0*, *GUSB*) and results were analyzed using the Delta-Delta $C_t$ method. We applied a linear regression model for gene expression level~log(APC concentration). To test for significance of the observed APC effects, we used the F-test of the linear regression model. We tested the residuals for normality using the Shapiro-Wilk test and for equal variance using the Bartlett test.

**In vitro static adhesion assay to assess the effects of recombinant sEPCR**. To quantify U937–HUVEC interactions, we used an in vitro static adhesion assay, as previously described[87]. U937 cells were seeded at a density of $1 \times 10^5$ cell/ml in T25 flasks and differentiated into macrophages in the presence of 100 ng/mL phorbol 12-myristate 13-acetate (PMA) for 48 h. HUVECs were seeded at a density of 27,174 cells/cm$^2$ (i.e., $1 \times 10^5$ cells per well of a 12-well plate) in 1 ml medium and left to attach overnight. HUVECs were then treated with 10 ng/ml TNF-α or vehicle control for 4 h. U937 cells were collected and re-suspended in fresh medium at a concentration of $1 \times 10^5$ cells cell/ml, and then incubated with various concentrations of anti-Mac-1 or recombinant human sEPCR (i.e., 0, 3, 6, 12 ng/ml). HUVEC monolayers (at ≥90% confluence) were rinsed in Phosphate NaCl (PBSA) buffer and incubated with 1 ml U937 cell suspension comprising $1 \times 10^5$ cells (±anti-Mac-1/sEPCR) at 37 °C for 5 min. After aspirating the U937 suspension, the HUVECs and any adherent U937 cells were gently rinsed four times in PBSA, and then a further 2 ml PBSA was added to the well. Using a phase-contrast video-microscope (Leica Microsystems, DMI3000B), pictures at 10-fold magnification were taken, choosing four different fields at random. Quantification of cell adhesion events was performed using the ImagePro v6.3 software. We applied a linear regression model for adhesion events~treatment concentration. To test for significance of the observed sEPCR and anti-Mac-1 effects, we used the *F*-test of the linear regression model. To test for significance of the IgG effect, we used a non-parametric linear model. We assessed the residuals for normality using the Shapiro-Wilk test and for equal variance using the Bartlett test. We also tested for a difference of slope coefficients between the IgG and anti-Mac-1 conditions by fitting the regression model with an interaction term (adhesion events~treatment concentration*condition). The P value was calculated using an ANOVA of the linear regression model.

**In vitro static adhesion assay to assess the effects of APC**. U937 cells were seeded and treated with PMA as described above. HUVECs and HCAECs were seeded at a density of 27,174 cell/cm$^2$ (i.e., $1 \times 10^5$ cells per well of a 12-well plate) in 1 ml medium and left to attach overnight. Endothelial cells were treated for 24 h with either: (i) 1 ng/ml TNF alone, (ii) 1 ng/ml TNF and 100 nM APC, or (iii) vehicle. U937 cells were then collected and re-suspended as above, but with no further treatments. Endothelial cell monolayers (at ≥90% confluence) were rinsed in PBSA buffer and incubated with 1 ml U937 cell suspension comprising $1 \times 10^5$ cells at 37 °C for 5 min. The monolayers were then rinsed, and adhesion events recorded and quantified as outlined above. We used paired one-tailed *t*-tests to test for differences in adhesion events between the TNF and TNF + APC conditions.

**Reporting summary**. Further information on research design is available in the Nature Research Reporting Summary linked to this article.

## Data availability

Genetic association data retrieved from the *PROCR*-219Gly phenome-scan are available through the UK Biobank ICD PheWeb (http://pheweb.sph.umich.edu/SAIGE-UKB/variant/20:33764554-A-G) and PhenoScanner (http://www.phenoscanner.medschl.cam.ac.uk/?query=rs867186&catalogue=GWAS&p=5e-8&proxies=None&r2=0.8&build=37/). GWAS summary statistics at the *PROCR* locus used for colocalization and Mendelian randomization analyses are available as follows: stroke (Malik 2018; https://www.ebi.ac.uk/gwas/publications/29531354), CAD (van der Harst 2017; https://www.ebi.ac.uk/gwas/publications/29212778) and APC (Sun 2018; https://www.ebi.ac.uk/gwas/publications/29875488) data are available for FTP download from the NHGRI-EBI Catalog of GWAS. FVII and PC data (Tang 2010) are available on request from the ARIC study at: https://sites.cscc.unc.edu/aric/distribution-agreements. Pulmonary embolism (phenotype ID: 20002_1093), VTE (I9_VTE) and DVT (20002_1094) data can be downloaded from the UK Biobank (http://www.nealelab.is/uk-biobank) using the following wget commands:

Pulmonary embolism: 'wget https://broad-ukb-sumstats-us-east-1.s3.amazonaws.com/round2/additive-tsvs/20002_1093.gwas.imputed_v3.both_sexes.tsv.bgz -O 20002_1093.gwas.imputed_v3.both_sexes.tsv.bgz'; VTE: 'wget https://broad-ukb-sumstats-us-east-1.s3.amazonaws.com/round2/additive-tsvs/I9_VTE.gwas.imputed_v3.both_sexes.tsv.bgz -O I9_VTE.gwas.imputed_v3.both_sexes.tsv.bgz'; DVT: 'wget https://broad-ukb-sumstats-us-east-1.s3.amazonaws.com/round2/additive-tsvs/20002_1094.gwas.imputed_v3.both_sexes.tsv.bgz -O 20002_1094.gwas.imputed_v3.both_sexes.tsv.bgz'. Supplementary Data 1 provides further information on the genetic data sources. All other data that support the findings of this study are available from the corresponding author upon reasonable request.

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

## Acknowledgements

We gratefully acknowledge the participation of all National Institute for Health Research (NIHR) BioResource Center Cambridge volunteers and thank the NIHR BioResource Center Cambridge and staff for their contribution. We thank the NIHR and NHS Blood and Transplant. We acknowledge Dr Honglin Song for her technical support. This work was supported by the British Heart Foundation (BHF) Cambridge Center of Excellence [RE/13/6/30180]. The Cardiovascular Epidemiology Unit is supported by core funding from the: UK Medical Research Council [MR/L003120/1], BHF [RG/13/13/30194; RG/18/13/33946] and NIHR Cambridge Biomedical Research Center [BRC-1215-20014]. The views expressed are those of the author(s) and not necessarily those of the NIHR or the Department of Health and Social Care. This work was further supported by BHF-DZHK VIAgenomics [SP/19/2/344612], a BHF Program Grant [RG/19/9/34655], the Van Geest Heart and Cardiovascular Diseases Research Fund and Health Data Research UK, which is funded by the UK Medical Research Council, Engineering and Physical Sciences Research Council, Economic and Social Research Council, Department of Health and Social Care (England), Chief Scientist Office of the Scottish Government Health and Social Care Directorates, Health and Social Care Research and Development Division (Welsh Government), Public Health Agency (Northern Ireland), BHF and Wellcome. D.S. is supported by a BHF Program Grant [RG/18/13/33946]. L.C was supported by a BHF Program Grant [RG/16/4/32218]. J.M.M.H. was supported by a BHF Program Grant [RG/13/13/30194] and the NIHR Cambridge Biomedical Research Center [BRC-1215-20014]. A.M. is funded by the NIHR Cambridge Biomedical Research Center [BRC-1215-20014] and the EU/EFPIA Innovative Medicines Initiative Joint Undertaking BigData@Heart [116074]. J.E.P. is supported by a UKRI Innovation Fellowship at Health Data Research UK (MR/S004068/2). M.S-L is supported by a Miguel Servet contract from the ISCIII Spanish Health Institute [CP17/00142] and co-financed by the European Social Fund. P.S. deVries is supported by American Heart Association grant number 18CDA34110116. C.S. is funded by the NIHR [NIHR133788] and the Medical Research Council [MR/P502091/1], and supported by the NIHR Cambridge Biomedical Research Center [BRC-1215-20014]. J.D. holds a BHF Professorship and a NIHR Senior Investigator Award. The US National Heart, Lung and Blood Institute (NHLBI) provided support for LITE via R01HL059367. This work was carried out in part using computing resources at the University of Minnesota Supercomputing Institute. The Atherosclerosis Risk in Communities (ARIC) study has been funded in whole or in part with US federal funds from the National Heart, Lung and Blood Institute (HHSN268201700001I, HHSN268201700002I, HHSN268201700003I, HHSN268201700004I, HHSN268201700005I, R01HL087641, R01HL086694); National Human Genome Research Institute (U01HG004402); and National Institutes of Health (HHSN268200625226C). The authors thank the staff and participants of the ARIC study for their important contributions. Infrastructure was partly supported by Grant Number UL1RR025005, a component of the National Institutes of Health and NIH Roadmap for Medical Research. The CHARGE Hemostasis Working Group acknowledges a NIH R01 HL134894 grant. SomaScan® and SOMAmer® reagent are registered trademarks of SomaLogic, Inc. This research has been conducted using the UK Biobank resource under Application Number 26865.

## Author contributions

D.S.P. and D.S. designed the study. D.S.P. and A.M.M. performed phenome-scans. L.C. performed Mendelian randomization analyses (with input from S. Burgess). L.C. and J.M.M.H performed statistical colocalization analyses. D.S. and D.S.P. conducted recall-study and analyzed data. S.M. and J.L. performed biomarker measurements as part of the recall-study. P.J.S. quantified EPCR levels on genotyped HUVECs (under guidance from N.J.S. and S.Y.). D.S. quantified EPCR levels on human monocytes and neutrophils (under guidance from H.M. and K.D.). D.S. performed RT-qPCR experiments. D.S. conducted leukocyte–endothelial cell adhesion assays (under guidance from N.F., C.S. and E.R.C.). A.D.G. conducted binding activity and competition assays of SOMAmer reagents (under guidance from D.J.S. and N.J.). S. Basu, J.S.P., W.T., N.P., M.S.-L., P. deVries and N.L.S. provided access to genetic datasets. D.S.P. and D.S. wrote the manuscript with input from L.C., J.E.P. and J.D. All authors reviewed and approved the final version of this manuscript.

## Competing interests

During the drafting of the manuscript, L.C. and J.M.M.H. became full-time employees of Novo Nordisk and D.S.P. became a full-time employee of AstraZeneca. J.E.P. has received travel and accommodation expenses and hospitality from Olink to speak at Olink-sponsored academic meetings. A.D.G., D.J.S. and N.J. are employees and stake-holders of SomaLogic. J.D. reports grants, personal fees and non-financial support from Merck Sharp & Dohme (MSD); grants, personal fees and non-financial support from Novartis; grants from Pfizer; and grants from AstraZeneca outside the submitted work. J.D. sits on the International Cardiovascular and Metabolic Advisory Board for Novartis (since 2010); the Steering Committee of UK Biobank (since 2011); the MRC International Advisory Group (ING) member, London (since 2013); the MRC High Throughput Science 'Omics Panel Member, London (since 2013); the Scientific Advisory Committee for Sanofi (since 2013); the International Cardiovascular and Metabolism Research and Development Portfolio Committee for Novartis; and the AstraZeneca Genomics Advisory Board (2018). The remaining authors declare no competing interests.

## Additional information

[1]British Heart Foundation Cardiovascular Epidemiology Unit, Department of Public Health and Primary Care, University of Cambridge, Cambridge, UK. [2]Department of Cardiovascular Sciences, University of Leicester, Leicester, UK. [3]National Institute for Health Research Leicester Biomedical Research Centre, University of Leicester, Leicester, UK. [4]Department of Genetics, Novo Nordisk Research Centre Oxford, Innovation Building, Old Road Campus, Roosevelt Drive, Oxford, UK. [5]Medical Research Council Biostatistics Unit, University of Cambridge, Cambridge, UK.

[6]Specialist Haemostasis Unit, Cambridge University Hospitals NHS Foundation Trust, Cambridge, UK. [7]Department of Haematology, University of Cambridge, Cambridge, UK. [8]National Health Service Blood and Transplant, Cambridge, UK. [9]National Institute for Health Research BioResource, University of Cambridge, Cambridge, UK. [10]Department of Medicine, University of Cambridge, Cambridge, UK. [11]Centre for Inflammatory Disease, Department of Immunology and Inflammation, Imperial College London, London, UK. [12]Health Data Research UK London, London, UK. [13]Division of Biostatistics, School of Public Health, University of Minnesota, Minneapolis, MN, USA. [14]Division of Epidemiology and Community Health, School of Public Health, University of Minnesota, Minneapolis, MN, USA. [15]Department of Laboratory Medicine and Pathology, School of Medicine, University of Minnesota, Minneapolis, MN, USA. [16]Genomics of Complex Diseases Group, Sant Pau Biomedical Research Institute, IIB-Sant Pau, Barcelona, Spain. [17]Cardiovascular Medicine Unit, Department of Medicine, Karolinska Institutet, Center for Molecular Medicine, Karolinska University Hospital, Stockholm, Sweden. [18]Human Genetics Center, Department of Epidemiology, Human Genetics, and Environmental Sciences; School of Public Health, The University of Texas Health Science Center at Houston, Houston, TX, USA. [19]Department of Epidemiology, School of Public Health, University of Washington, Seattle, WA, USA. [20]Seattle Epidemiologic Research and Information Center, Department of Veterans Affairs Office of Research and Development, Seattle, WA, USA. [21]Kaiser Permanente Washington Health Research Institute, Seattle, WA, USA. [22]SomaLogic Inc, Boulder, CO, USA. [23]National Heart and Lung Institute, Imperial College London, London, UK. [24]British Heart Foundation Centre of Research Excellence, University of Cambridge, Cambridge, UK. [25]National Institute for Health Research Blood and Transplant Research Unit in Donor Health and Genomics, University of Cambridge, Cambridge, UK. [26]Health Data Research UK Cambridge, Wellcome Genome Campus and University of Cambridge, Cambridge, UK. [27]Department of Human Genetics, Wellcome Sanger Institute, Hinxton, UK. [68]These authors contributed equally: David Stacey, Lingyan Chen. *A list of authors and their affiliations appears at the end of the paper. ✉email: dsp35@medschl.cam.ac.uk

## CHARGE Hemostasis Working Group

Abbas Dehghan[28], Adam S. Heath[18], Alanna C. Morrison[18], Alex P. Reiner[19], Andrew Johnson[29], Anne Richmond[30], Annette Peters[31], Astrid van Hylckama Vlieg[32], Barbara McKnight[33], Bruce M. Psaty[34], Caroline Hayward[30], Cavin Ward-Caviness[35], Christopher O'Donnell[36], Daniel Chasman[37], David P. Strachan[38], David A. Tregouet[39], Dennis Mook-Kanamori[32], Dipender Gill[28], Florian Thibord[29], Folkert W. Asselbergs[40], Frank W. G. Leebeek[41], Frits R. Rosendaal[32], Gail Davies[42], Georg Homuth[43], Gerard Temprano[16], Harry Campbell[44], Herman A. Taylor[45], Jan Bressler[18], Jennifer E. Huffman[46], Jerome I. Rotter[45], Jie Yao[45], James F. Wilson[47], Joshua C. Bis[34], Julie M. Hahn[18], Karl C. Desch[48], Kerri L. Wiggins[34], Laura M. Raffield[49], Lawrence F. Bielak[50], Lisa R. Yanek[51], Marcus E. Kleber[52], Maria Sabater-Lleal [ID][16,17], Martina Mueller[31], Maryam Kavousi[53], Massimo Mangino[54], Matthew P. Conomos[33], Melissa Liu[29], Michael R. Brown[18], Min-A Jhun[50], Ming-Huei Chen[55], Moniek P. M. de Maat[41], Nathan Pankratz [ID][15], Nicholas L. Smith[19,20,21], Patricia A. Peyser[50], Paul Elliot[28], Paul S. de Vries[18], Peng Wei[56], Philipp S. Wild[57], Pierre E. Morange[58], Pim van der Harst[59], Qiong Yang[60], Ngoc-Quynh Le[16], Riccardo Marioni[61], Ruifang Li[32], Scott M. Damrauer[62], Simon R. Cox[63], Stella Trompet[64], Stephan B. Felix[65], Uwe Völker[43], Weihong Tang[14], Wolfgang Koenig[66], J. Wouter Jukema[67] & Xiuqing Guo[45]

[28]Department of Epidemiology and Biostatistics, School of Public Health, Imperial College London, London, UK. [29]National Heart Lung and Blood Institute, Division of Intramural Research, Population Sciences Branch, The Framingham Heart Study, Framingham, MA, USA. [30]Medical Research Council Human Genetics Unit, Institute of Genetics and Molecular Medicine, University of Edinburgh, Western General Hospital, Edinburgh, UK. [31]Research Unit Molecular Epidemiology, Helmholtz Zentrum München, München, Germany. [32]Department of Clinical Epidemiology, Leiden University Medical Center, Leiden, The Netherlands. [33]Department of Biostatistics, University of Washington, Seattle, WA, USA. [34]Cardiovascular Health Research Unit, Department of Medicine, University of Washington, Seattle, WA, USA. [35]Office of Research and Development, US Environmental Protection Agency, Chapel Hill, NC, USA. [36]Cardiology, VA Boston Healthcare System, Boston, MA, USA. [37]Division of Preventive Medicine, Brigham and Women's Hospital, Boston, MA, USA[38]Population Health Research Institute, St George's University of London, London, UK. [39]Bordeaux Population Health Research Center, University of Bordeaux, Bordeaux, France. [40]Department of Cardiology, Division of Heart and Lungs, University Medical Center Utrecht, Utrecht University, Utrecht, The Netherlands. [41]Department of Hematology, Erasmus MC University Medical Center, Rotterdam, The Netherlands. [42]Lothian Birth Cohorts, Department of Psychology, University of Edinburgh, Edinburgh, UK. [43]Department of Functional Genomics, University Medicine Greifswald, Greifswald, Germany. [44]Global Health Research, Usher Institute for Population Health Sciences and Informatics, University of Edinburgh, Edinburgh, UK. [45]The Institute for Translational Genomics and Population Sciences, Department of Pediatrics, The Lundquist Institute for Biomedical Innovation at Harbor-UCLA Medical Center, Torrance, CA, USA. [46]Massachusetts Veterans Epidemiology Research and Information Center (MAVERIC), VA Boston Healthcare System, Boston, MA, USA. [47]Medical Research Council Human Genetics Unit, Institute of Genetics and Cancer, University of Edinburgh, Western General Hospital, Edinburgh, UK. [48]Department of Pediatrics, University of Michigan, CS Mott Children's Hospital, Ann Arbor, MI, USA. [49]Department of Genetics, University of North Carolina at Chapel Hill, Chapel Hill, NC, USA. [50]Department of Epidemiology, School of Public Health, University of Michigan, Ann Arbor, MI, USA. [51]Department of Medicine, Johns Hopkins University School of Medicine, Baltimore, MD, USA. [52]SYNLAB MVZ für Humangenetik Mannheim, Mannheim, Germany. [53]Department of Epidemiology, Erasmus Medical Center, University Medical Center Rotterdam, Rotterdam, The Netherlands. [54]Department of Twin Research and Genetic Epidemiology, Kings College London, London, UK. [55]Population Sciences Branch, National Heart, Lung, and Blood Institute, Framingham, MA, USA. [56]Department of Biostatistics, The University of Texas MD Anderson Cancer Center, Houston, TX, USA. [57]Department of Cardiology, Cardiology I, University Medical Center, Johannes Gutenberg University Mainz, Mainz, Germany. [58]Hematology Laboratory, La Timone

University Hospital of Marseille, Marseille, France. [59]Department of Cardiology, University Medical Center Utrecht, Utrecht, The Netherlands. [60]Department of Biostatistics, Boston University School of Public Health, Boston, MA, USA. [61]Centre for Genomic and Experimental Medicine, Institute of Genetics and Molecular Medicine, University of Edinburgh, Edinburgh, UK. [62]Department of Surgery, Perelman School of Medicine, University of Pennsylvania, Philadelphia, PA, USA. [63]Department of Psychology, University of Edinburgh, Edinburgh, UK. [64]Section of Gerontology and Geriatrics, Department of Internal Medicine, Leiden University Medical Center, Leiden, The Netherlands. [65]Department of Internal Medicine B, University Medicine Greifswald, Greifswald, Germany. [66]DZHK (German Centre for Cardiovascular Research), Partner Site Munich Heart Alliance, Munich, Germany. [67]Department of Cardiology, Leiden University Medical Center, Leiden, The Netherlands.

