## [Peer Review File · Nature Communications]

Elucidating mechanisms of genetic cross-disease associations at the PROCR vascular disease locusReviewers' Comments:

Reviewer #1:

Remarks to the Author:

This article by Stacey and colleagues investigates the health and functional effects of a non-synonymous variant in the PROCR gene, which has been shown to be associated with a lower CAD risk, but a higher VTE risk in genome-wide associated studies. This paper is clear, straightforward, very well written and most of the conclusions are supported by the results. The authors of this paper used a highlight translational approach.

Using two approaches, one combining >1400 disease-related traits in the UK Biobank and another approach using more targeted phenotypes associated with traits of interest, they confirmed the association of this variant with lower CAD risk and higher VTE risk. This approach also suggested and association with stroke as well as tissue factors and the levels of proteins involved in the protein C pathway. The identification of shared genetic etiology (i.e. the lead SNPs associated with phenotypes being the same SNPs linked with circulating proteins) is an innovative way to conclude this section. A discussion around the extent of population overlapping between the studies that have identified this variant and the phenome-scan here would be welcome. The legend of figure 2 is too long. Most of the details presented there should be presented in the materials and methods section.

The state-of-the-art bi-directional Mendelian randomization set of analysis also nicely supports the causal role of protein C as a protective factor for CAD and as a causative factor for VTE. This was performed by the use of strong and independent ci-acting variants at the PROC locus. The authors mention that they used a 1MB window to identify independent SNPs but it is unclear what was the window used to identify these SNPs, 200kb, 500 kb, 1Mb within the PROC gene? The authors did not justify the use of the simple median and weighted median MR methods in the methodology section. It is also unknown if outliers were identified using MR-PRESSO. It would be helpful to understand how the authors calculated the F-statistic for instrument strength.

The recall by genotype approach is also an important validation step. How was the study sample size determined? Was the SNP directly typed by the array or was it imputed?

In the results section, the authors claim that "endothelial cells represent the primary source of sEPCR". However, here they only compared endothelial cells to monocytes and differentiated endothelial cells. This claim does not appear to be supported by the results, unless they provide evidence that sEPCR is not secreted by other endocrine organs such as adipose tissue, liver, etc. This should be rephrased. Also, in this section, the authors mention that they are studying elevated protein C "due to rs867186". Since genotype was not considered in the functional assays, these results may be due to any event that may lead to higher sEPCR, so this section should be rephrased accordingly.

More information is needed to show that the volume of cells were taken by flow cytometry were consistent across experiments.

The cell-adhesion experiments are an interesting addition to the paper. This is however the weakest part of the paper. Here, the authors suggest that variation in PROCR might influence CAD risk because of potential differences in leukocyte-endothelial cell adhesion. However, these results are not very convincing. First the effect of APC is small and it barely reached the level of statistical significance. It is also unclear how statistical significance was inferred here as the methods are not described. Were parametric or non-parametric used? Analyses of variance? It is also unknown if this experiment was performed more than once. The "adhesion events" also seems like a rather vague study outcome, and here it was the only outcome chosen to reach the conclusion that APC influences cell-adhesion. Were cell adhesion events determined automatically or by the judgement of study investigators? If the latter, how many investigators documented cell adhesion events and were they blinded to the study hypothesis?

Did the authors assess cell viability throughout these experiments?

The authors refer to other studies suggesting that endothelial cells exposed to APC have lower levels of expression of adhesion molecules such as ICAM-1 and VCAM-1. Could these molecules influence the apparently lower cell-adhesion observed in the presence of APC in this study?

The materials and methods section should be rearranged so that the methods for each part of the investigations is presented in a chronological order (i.e., begin with the phenome scan and finish by the functional experiments).

Reviewer #2:

Remarks to the Author:

The manuscript by Stacey et al. presents an investigation of an observation for the pleiotropic genetic variant rs867186 A>G (S219G) in the PROCR gene (endothelial protein C receptor, EPCR) which has a dual, opposing effects on coronary artery disease (CAD, lower risk) and venous thromboembolism (VTE, higher risk). The background and the rationale of the study are presented in a well-structured manner and the investigation of this cross-disease association is deemed to be of high importance. However, there are several major concerns with the experiments aiming to elucidate the mechanistic basis behind 219G association with lower CAD risk and higher VTE risk:

- As pointed out in the Introduction, the association of the 219G genotype with CAD and VTE is already known, and the analysis is essentially confirmatory.
- The most interesting aspect is the analysis of the molecular mechanism behind these seemingly opposing effects. Unfortunately, the data to support any sort of mechanism is only superficial and definitely not conclusive.

Other major issues:

PC/APC, EPCR/sEPCR levels and coagulation

- Figure 2C demonstrates that protein C (PC) is elevated in the 219G genotype shown by all ELISA, SOMAscan and chromogenic assay. What is the mechanistic explanation for that? How will the EPCR missense variant S219G lead to increased PC levels? Why are the levels of PC not measured by an immunoassay in the recall-by-genotype study? The finding that PC levels/activation is increased in the 219G genotype do not match the notion that this allele is associated with higher thrombosis (i.e. VTE) as the opposite will be expected.
- The findings about activated PC (APC) levels in 219G genotype are discrepant: APC was shown to be upregulated by SOMAscan but not by immunoassay (Figure 2C and Figure 5). 219G genotype exhibits an elevated soluble EPCR (sEPCR) and even a "truncated" EPCR according to the Introduction – this "dysfunctional" EPCR will be expected to decrease APC formation, why is it increased or unchanged instead? APC is an anticoagulant – why its increase will be associated with higher thrombotic potential?
- "To confirm the specificity of the binding events, we measured the binding activity of the PC and APC SOMAmer reagents to a range of relevant proteins, specifically, PC, APC, sEPCR, thrombin, FV, FVIIa, Protein S and thrombomodulin". None of these other protein associations were independently confirmed by immunoassays. Given the observed discrepancy of the SOMAmer measurements with immunoassay for activated protein C, this should be done.
- The SOMAmer assay provides measurements of hundreds or thousands of proteins. The associations of the minor (G) allele of rs867186 (PROCR-219Gly) with all measurements should be presented.
- Previous studies have shown that physiologically high sEPCR does not affect the anticoagulation potential of the protein C pathway as only the membrane EPCR (mEPCR) is important for APC

formation and anticoagulation. An increase of APC will therefore be due to higher mEPCR which together with the higher sEPCR will be indicative of higher total levels of EPCR in 219G genotype. Authors should specifically investigate the mEPCR in their in vitro studies – is the EPCR expression changed in 219G genotype? Are the mEPCR levels changed in 219G genotype? Are the sEPCR levels elevated at the expense of a reduced mEPCR? Authors should remove the diagram showing reduction in mEPCR on Figure 7 since this has not been demonstrated in the study (or otherwise investigate it).

- EPCR can also be expressed on platelets (especially activated platelets). Blood processing in this study does not ensure that plasma is free of platelets: firstly, spinning blood at 4 deg would likely lead to platelet activation, and secondly, 1,000g for 15 min will not provide a platelet-free plasma. What is the evidence that EPCR measured in plasma is soluble and not platelet-associated?
- FVII is also elevated in the 219G genotype. What is the mechanistic explanation for that? How is this affecting protein C pathway? For instance, FVII is a known ligand for EPCR where it can displace PC and decrease its activation, thus leading to a procoagulant effect. Does the missense S219G variant affect FVII binding to EPCR? FVII and its interactions with EPCR have not been studied at all while there is a very vague argument about its involvement in the Discussion which is hardly supported by any data.
- Authors mention that FVII-EPCR interaction leads to clearance by endocytosis. Hence, 219G genotype should have more EPCR endocytosis due to higher FVII being present. At the same time authors show higher sEPCR in 219G – these are two contradicting arguments.
- There is also a speculation about the increased D-dimer in the Discussion which is similarly vague and unproven. Authors should remove the diagram showing fibrin deposition increase in 219G genotype on Figure 7 unless they specifically investigate this (e.g. fibrin formation assay).
- An important thing which is missing is a coagulation assay to show whether plasma from 219G genotype indeed has higher propensity to form clots.

Flow cytometry

- EPCR expression is 75% higher in HUVECs compared to monocytes – how is this calculated? The use of the blocking antibody demonstrates specificity but its use to estimate the level of expression on different cell types is difficult to understand. Despite the fact that primary monocytes/neutrophils show lower signal for EPCR on Figure S6, practically all of the cells are positive for EPCR. Authors are encouraged to investigate the difference in EPCR expression using an alternative method as well.
- Monocytes express Fc receptors and therefore incorporation of an Fc receptor blocking antibody control is desirable to show specificity of the signal.

Inflammation/CAD

- The leukocyte-endothelial cell adhesion assay is poorly designed, and the conclusions are purely speculative. Firstly, this is a static assay performed by addition of macrophages on top of a monolayer of HUVECs. However, authors state that their assay is performed as previously described in two articles (ref 77 and 78) which conduct an adhesion assay under flow. Rolling, arresting and transmigration cannot be examined using a static assay as authors say they did. Rolling and arresting occur under flow conditions, while transmigration requires the cells to extravasate in another basal compartment, which is not possible when the HUVECs are seeded in a 2D tissue culture dish. What was quantified exactly in this model and how? Cells are used at $\geq 90\%$ confluency so how do authors ensure macrophages do not bind to the 10% of the tissue culture plastic? Furthermore, washing the cells “gently” with saline is likely to be highly inconsistent. Usually static assays are performed by inverting a plate and centrifuging it to remove non-adherent cells or similar. A flow assay should be performed to confirm that there is indeed an effect of APC on adhesion or transmigration of macrophages.
- HUVECs are not a suitable model for the research question of this study – these are foetal venous cells which are not related to atheromatous formation in coronary arteries. Arterial cells should be used, e.g. coronary artery endothelial cells, in order to provide robust evidence for effects in CAD. This is especially important for this study where CAD and VTE are compared.
- Which surface adhesion molecules are reduced by APC treatment? Authors should show this by e.g. qPCR and western blotting. Since APC is only present during the cell co-incubation and the adhesion assay only lasts 5 min, how is it possible for APC to induce any transcriptional or translational

changes?

- A single finding of an in vitro adhesion assay which was not designed well cannot be generalised to indicate effects on atherosclerosis, a complex multifactorial disease involving multiple cell types and risk factors. Such conclusions can only be made if multiple models are used – e.g. use of diseased cells, use of cells with 219G genotype, use of cells overexpressing EPCR or with downregulation of EPCR, murine model of atherosclerosis to show that 219G indeed leads to increased atheromatous lesion formation etc.
- The authors' conclusions about the findings of the in vitro adhesion assay are an overinterpretation of the data. There is no anti-inflammatory role of APC shown in this manuscript and this statement should be removed. The hypothetical activation of PAR-1 by APC and reduction of ICAM-1 and VCAM-1 explained in the Discussion is speculative and practically impossible to occur in the adhesion assay – APC cannot have transcriptional effects on the endothelial cells in a 5-min assay. The speculation about APC-PAR-1 interaction, the statement that APC has a protective role in atherosclerosis and the diagram showing PAR-1 signalling and reduction of vascular inflammation on Figure 7 should all be removed or specifically investigated prior to inclusion.
- The statement at the bottom of page 9 that 219G genotype has lower susceptibility to atherosclerosis due to increased APC levels contradicts the findings on Figure 5 which show no APC elevation in 219G genotype.

Minor issues:

- Methods state that both plasma and serum have been collected. Which are the experiments performed with serum samples? What was the rationale behind this?
- Please correct cell seeding densities – cell density should be expressed as cell/cm² for adhesion cells or cells/ml for cells in suspension.
- Please use the same scale for X-axes on Figures S4, S5 and S6 and make the graphs equal in size.
- Please state in the Methods section the concentration of sEPCR, APC and Mac-1 used in the adhesion assay. Please include details of the timing of addition of sEPCR, APC and Mac-1.

Reviewer #3:

Remarks to the Author:

The authors eloquently present the results from a compelling series of experiments that dissect the pleiotropic and seemingly paradoxical effects of the PROCRA-p.S129G variant on CAD and VTE. They use a combination of approaches from molecular epidemiology, statistical genetics, recall-by-genotype deep phenotyping, and cell culture assays to tease apart the mechanisms underlying the pleiotropy at this variant. The statistical analyses are well conceptualized, appropriately conducted and reported, and clearly presented, as is the translational and wet-bench science. The work critically advances our understanding of the locus and identifies the need for more detailed mechanistic studies. Nonetheless, I do have the following critiques:

1. Is sEPCR targeted by the SOMAScan platform and if so, are levels associated with rs867186 in data from KORA and/or INTERVAL and does the genetic signal in that region colocalize with CAD, VTE, PC, and APC?
2. The authors clearly demonstrate the statistical evidence of the rs867186 variant being causal for PC, APC, and FVIII levels, and DVT and CAD outcomes, as well as nice statistical evidence of PC and APC levels being causal for CAD and VTE. In the later experiment they used a series of independent variants in PROCRA to instrument the exposure. Is it possible to further refine these analyses to more directly demonstrate a causal pathway from rs867186 to PC/APC to outcomes using a mediation-based approach?
3. The authors performed MR for PC and APC effects on cardiovascular outcomes and then reverse MR to exclude bidirectional effects. It seems odd that Table 1 has the Forward MR for PC and the Reverse MR for APC. In the absence of a compelling reason for this it might be more appropriate to have the table reflect either PC or APC and then have the other species in the supplement.

4. The authors are to be commended for the inclusion of recall-by-genotype analyses and I am in full support of the approach they present in Figure 1; it is great to see it in practice. Their discussion of the generalizability of the approach they use in the paper is spot on and well taken. The manuscript would benefit, however, from additional context about what the authors feel recall-by-genotype analyses added to this specific study beyond what is already known or otherwise presented in the manuscript. As the authors explain in the introduction, the PROCR-p.S129G has already been well characterized and is known to result in increased sEPCR. The increased PC activity, which is essentially a proxy for PC levels, is clearly shown earlier in the paper. And they do not detect an effect of genotype on the TAT complex or APC in the recall-by-genotype analyses.

5. How do the authors explain the apparent discrepancy between the epidemiological analyses and recall-by-genotype based analyses with respect to the rs867186 – APC relationship?

6. The putative mechanistic pathway that the authors propose for PROCR to increase VTE risk seems to be centered on increased sEPCR resulting in decreased FVII, likely through decreased membrane EPCR. The evidence here seems to come from pre-existing literature, the recall -by-genotype studies showing increased sEPCR in rs867186 carriers, and the epidemiological association of rs867186 with decreased FVII levels. Is there evidence either in the literature, or that the authors can provide experimentally, that demonstrates that rs867186 is, in fact, associated with decreased surface levels of EPCR and thus link the increased levels of sEPCR with decreased FVII mechanistically, rather than just hypothesize that increased sEPCR results in decreased membrane EPCR?

7. Additionally, the pathway the authors posit for rs867186 leading to VTE does not seem to take into account the MR experiments suggesting that PC and APC are causal for these outcomes. How do they reconcile the results of the MR experiments with their proposed mechanism which appears to be independent of actual PC/APC levels?

8. The manuscript would be improved by presenting the putative mechanistic chain with a bit more circumspection. There are many steps in Figure 7, especially with respect to VTE, that are not clearly demonstrated in the paper and remain hypotheses. These steps in the mechanistic chain will likely require experiments using genetically edited iPSC, genotype specific cellular assays, or model systems to test. Understandably these experiments are most likely beyond the scope of this paper but I think more explicitly recognizing the remaining uncertainty, and where in the pathway it lies, would make the paper stronger.

Elucidating mechanisms of genetic cross-disease associations: an integrative approach implicates protein C as a causal pathway in arterial and venous diseases

Rebuttal letter

Reviewer #1 (Remarks to the Author):

1. This article by Stacey and colleagues investigates the health and functional effects of a non-synonymous variant in the PROC gene, which has been shown to be associated with a lower CAD risk, but a higher VTE risk in genome-wide associated studies. This paper is clear, straightforward, very well written and most of the conclusions are supported by the results. The authors of this paper used a highlight translational approach.

We thank the reviewer for the positive feedback on our manuscript.

2. Using two approaches, one combining >1400 disease-related traits in the UK Biobank and another approach using more targeted phenotypes associated with traits of interest, they confirmed the association of this variant with lower CAD risk and higher VTE risk. This approach also suggested an association with stroke as well as tissue factors and the levels of proteins involved in the protein C pathway. The identification of shared genetic etiology (i.e. the lead SNPs associated with phenotypes being the same SNPs linked with circulating proteins) is an innovative way to conclude this section. A discussion around the extent of population overlapping between the studies that have identified this variant and the phenome-scan here would be welcome.

The phenome-scan shown in Fig. 2A was based exclusively on the UK Biobank data. In the targeted phenome-scan of the circulatory system shown in Figs. 2B and C, we retrieved the largest available genetic association dataset. The datasets that report a significant association of rs8671786 with disease outcomes were either derived from UK Biobank itself (VTE, DVT); from a meta-analysis of GWAS that included UK Biobank and cohorts of CARDIoGRAMplusC4D (CAD); or were independent of UK Biobank (VTE [INVENT consortium] and stroke phenotypes [MEGASTROKE consortium]). For CAD, a potential sample overlap between the UK Biobank and cohorts of CARDIoGRAMplusC4D was estimated to be <0.1%, and no evidence was found that this biased the test statistics (van der Harst et al (2018), Circ Res 122, 433-443). All intermediate traits related to the cardiovascular system were derived from datasets that were independent of UK Biobank, except for the haematological traits, which showed null associations with rs867186. Data on biomarkers that were measured on different analytical platforms (e.g. factor VII and protein C) were derived from individuals from independent cohorts. However, since we do not have full access to the individual-level data, we cannot fully exclude the possibility of a small sample overlap across these cohorts.

3. The legend of figure 2 is too long. Most of the details presented there should be presented in the materials and methods section.

In the revised manuscript, we have shortened the legend for Fig. 2 and moved most of the details to the Methods section, as suggested by the reviewer.

4. The state-of-the-art bi-directional Mendelian randomization set of analysis also nicely supports the causal role of protein C as a protective factor for CAD and as a causative factor for VTE. This was performed by the use of strong and independent ci-acting variants at the PROC locus. The authors mention that they

used a 1MB window to identify independent SNPs but it is unclear what was the window used to identify these SNPs, 200kb, 500 kb, 1Mb within the PROC gene?

The instrumental variables (IVs) for protein C were derived from a trans-pQTLs at the *PROC* locus. The *PROC* region [chr20: 31,916,110–35,505,723; hg19] was defined based on recombination rates, as reported in Howson et al. (2017) Nature Genet. 49, 1113-1119. To identify the approximately independent variants for the *PROC* region, we performed LD clumping within the region by PLINK v1.90b. We have included these additional details to the Methods section of the revised manuscript.

5. The authors did not justify the use of the simple median and weighted median MR methods in the methodology section. It is also unknown if outliers were identified using MR-PRESSO. It would be helpful to understand how the authors calculated the F-statistic for instrument strength.

We applied several MR methods to derive the causal estimates. The rationale is that if all methods come to similar conclusions, the causal estimates are more likely to be true as these methods have different underlying assumptions. The standard inverse-variance weighted (IVW) method assumes that all genetic variants are valid IVs. The median-based methods have greater robustness to individual genetic variants with strongly outlying causal estimates compared with the inverse-variance weighted and MR-Egger methods. Formally, the simple median method gives a consistent estimate of the causal effect when at least 50% of the genetic variants are valid instrumental variables. The weighted median method assumes that a majority of genetic variants are valid IVs. Suppl. Table 3 provides a summary of the selected instrumental variables for Mendelian randomization analyses.

Applying MR-PRESSO, we identified one outlier (rs112928119) when estimating causal relationship with PE/VTE/DVT. The MR-PRESSO outliers have now been added to the Suppl. Table 4 of the revised manuscript.

The F-statistics for each instrumental variable were derived from: $F = \beta^2 / SE^2$ (as described in Palmer et al. (2012) Stat Methods Med Res. 21, 223-242). We have included these details in the Methods section.

6. The recall by genotype approach is also an important validation step. How was the study sample size determined? Was the SNP directly typed by the array or was it imputed?

For our recall-study, we recruited all available participants of the NIHR Cambridge BioResource who carried the minor allele of rs867186 and satisfied the pre-specified inclusion criteria (n=18). Eligible participants who were heterozygous or homozygous for the major allele were then matched with respect to sex and age.

Based on the available sample cohort, we performed power calculations to assess the required effect size to observe a signal with 80% statistical power and a type I error rate of $\alpha=0.05$. The results of the calculations are shown below and have been added to the revised manuscript as Suppl. Fig. 9.

We note that it is challenging to estimate the actual effect size of the measured biomarkers by using the effect size from previous population-based studies, because the application of a stricter control of experimental conditions and the matching of phenotypic variables of participants (e.g. sex, age, BMI, food intake prior to venepuncture), as well as different normalization/transformation procedures, can have a considerable impact on the variability and estimated effect size of the measurement.

We can confirm that rs867186 was directly typed by the array used for the NIHR BioResource panel of healthy volunteers.

Suppl. Fig. 9. Power calculations for the recall-by-genotype study. We compared the statistical power for varying standardised per-allele effect sizes (β) in a cohort consisting of an equal number of major and minor homozygotes of a genetic variant of interest. The calculations are independent of the allele frequency of the variant. The x-axis is the total sample size of the recall experiment; the number of eventually recruited homozygotes ($n=36$) is indicated with a dashed line. The y-axis is the statistical power; 80% is indicated with a dashed line. We assume a type I error rate of $\alpha=0.05$. The power calculation is for an equal variance two-sample two-tailed t-test, as further described in Corbin et al. (2018) *Nature Commun.* 9, 711.

- In the results section, the authors claim that “endothelial cells represent the primary source of sEPCR”. However, here they only compared endothelial cells to monocytes and differentiated endothelial cells. This claim does not appear to be supported by the results, unless they provide evidence that sEPCR is not secreted by other endocrine organs such as adipose tissue, liver, etc. This should be rephrased. Also, in this section, the authors mention that they are studying elevated protein C “due to rs867186”. Since genotype was not considered in the functional assays, these results may be due to any event that may lead to higher sEPCR, so this section should be rephrased accordingly.

We fully agree with the reviewer and have removed the statement that endothelial cells represent the primary source of sEPCR.

We have now conducted additional experiments that directly assessed the impact of rs867186 on EPCR levels using genotype-specific human umbilical vein endothelial cells (HUVECs). We detected lower levels of membrane-bound EPCR in HUVECs obtained from homozygotes of the rs867186-G-allele compared to homozygotes of the A-allele. In HUVECs treated with phorbol 12-myristate 13-acetate (PMA), a potent agent to enhance ectodomain shedding, we also found lower levels in homozygotes of the G-allele compared to homozygotes of the A-allele. Taken together, these findings are consistent with

higher rates of EPCR shedding from endothelial cells in carriers of *PROCR*-rs867186-G. The results of these experiments are shown below and have been added to the revised manuscript as Figs. 4B and C.

Figure 4. Effect of rs867186 genotype on plasma biomarkers and EPCR expression on HUVECs. (A) Boxplots showing the distribution of the plasma biomarker levels as a function of rs867186 genotype in up to 52 individuals. We measured plasma levels of sEPCR, APC and TAT complex using immunoassays, and PC levels using a chromogenic assay. All measurements were done with three technical replicates. Bold lines and boxes represent the median and interquartile range of the data, respectively. Dashed lines indicate the fitted linear regression model for biomarker-genotype. P-values for the additive regression model are indicated. **(B)** Boxplots showing the distribution of mEPCR levels on HUVECs obtained from homozygotes of the rs867186-G-allele or A-allele ($n=3$ cell lines per individual). Data show mean intensity values of mEPCR on untreated HUVECs, normalized to mean intensity values of homozygotes of the rs867186-A-allele. P-values were calculated using a one-tailed t-test. **(C)** Boxplots showing the distribution of mEPCR levels on HUVECs obtained from homozygotes of the rs867186-G-allele or A-allele ($n=3$ cell lines per individual). Data show mean intensity values of mEPCR on HUVECs simulated with DMSO (vehicle control) or 50 nM phorbol 12-myristate 13-acetate (PMA), normalized to mean intensity values of homozygotes of the rs867186-A-allele. P-values were calculated using a paired one-tailed t-test. All experiments were performed with three technical replicates per cell line. EPCR expression levels were quantified using flow cytometric analysis (**Methods**). Bold lines and boxes represent the median and interquartile range of the data, respectively.

8. More information is needed to show that the volume of cells taken by flow cytometry were consistent across experiments.

Across all of our experiments, cells were resuspended to a final concentration of 10^5 cells (counting was performed using a haemocytometer) per 100 μ l in a total volume of 500 μ l and were stained using 0.125 μ g EPCR antibody per 100 μ l. Since we performed flow cytometry using FACS-lysed blood (comprised of heterogeneous cell types) and two different cultured cell lines (i.e. HUVECs and U937 cells), we did not work to ensure a consistent number of acquisitions between experiments (i.e. between Supplementary Figures). However, we did ensure a minimum of 5,000 events across all experiments, but note that after gating our analyses could sometimes be based on fewer than 5,000 events (e.g. Fig. S7B). In each of our overlay plots (Figs. S4-S7), we have plotted the median fluorescence intensity (MFI), which is robust to

differences in cell number between experiments/conditions. We have also clearly indicated the cell numbers used to compute the MFI for each condition underneath each of our overlay plots.

9. The cell-adhesion experiments are an interesting addition to the paper. This is however the weakest part of the paper. Here, the authors suggest that variation in PROCR might influence CAD risk because of potential differences in leukocyte-endothelial cell adhesion. However, these results are not very convincing. First the effect of APC is small and it barely reached the level of statistical significance. It is also unclear how statistical significance was inferred here as the methods are not described. Were parametric or non-parametric used? Analyses of variance? It is also unknown if this experiment was performed more than once.

We agree with the reviewer that the cellular effects initially observed in the adhesion assays were subtle and required further validation. To this end, we have completely revised this section and performed additional experiments. We conducted (i) RT-qPCR experiments to assess the effect of APC on key cellular adhesion molecules; (ii) additional adhesion assays with multiple biological (i.e. endothelial cells obtained from 3 independent donors) and technical (3 experiments for each of the 3 donors) replicates. Furthermore, we performed the RT-qPCR experiments and adhesion assays using two endothelial cell models: human umbilical vein endothelial cells (HUVECs) and human coronary artery endothelial cells (HCAECs).

The results from these new experiments are reported in section “Effect of APC on cell adhesion molecule expression and leukocyte–endothelial cell adhesion” of our revised manuscript, and shown below and in Fig. 5 of the revised manuscript. In brief, we found that APC attenuates the TNF-induced effect on *ICAM1* mRNA expression levels in both HUVECs and HCAECs, and that APC also attenuates U937 cell adhesion to endothelial monolayers in both cell models. For these new experiments, we altered the concentration of TNF used to stimulate endothelial cells from 10 ng/ml to 1 ng/ml. This is because when optimizing our RT-qPCR experiments, we found that 1 ng/ml TNF enabled more reliable and stable detection of an effect of APC on *ICAM1* mRNA levels. We hypothesise that 10 ng/ml TNF may have had a dominant cellular response that masked the comparatively subtle effect of APC in our original adhesion assays.

As a result of the revised experimental design, we have also amended the assessment of statistical significance for the *in vitro* static adhesion assays. For assays evaluating the effects of APC, we used paired one-tailed t-tests to test for differences in adhesion events between the TNF and TNF+APC conditions. Data were tested for normality using the Shapiro-Wilk test and for equal variance using the F-test. For assays testing the effects of recombinant sEPCR, we applied a linear regression model for adhesion events~treatment concentration. To test for significance of the observed sEPCR and anti-Mac-1 effects (Suppl. Fig. 8), we used the F-test of the linear regression model. To test for significance of the IgG effect, we used a non-parametric linear model. We assessed the residuals for normality using the Shapiro-Wilk test and for equal variance using the Bartlett test.

Figure 5. Effect of APC on cell adhesion molecule expression and leukocyte–endothelial cell adhesion. (A) Barplots showing gene expression levels of ICAM1 and PROCR in human umbilical vein endothelial cells (HUVECs) and human coronary artery endothelial cells (HCAECs) relative to the control condition (i.e. 0 nM APC; indicated with a dashed line). Cells were co-incubated with 1 ng/ml TNF- α and varying concentrations of APC (0, 0.1, 1, 10, 100 nM) for 24 hr. Error bars show standard deviations of the means. The blue and green lines indicate the fitted linear regression model for gene expression level–log(APC concentration). P-values for the F-test of the linear regression model are shown. **(B)** Barplots showing mean cell adhesion events using static adhesion assays with PMA-stimulated monocytic cells (U937) and TNF- α -activated HUVECs or HCAECs. Cells were co-incubated with 1 ng/ml TNF- α and 10 nM APC for 24 hr (**Methods**). Error bars show standard deviations of the means. P-values were calculated using paired one-tailed t-tests.

10. The “adhesion events” also seems like a rather vague study outcome, and here it was the only outcome chosen to reach the conclusion that APC influences cell-adhesion. Were cell adhesion events determined automatically or by the judgement of study investigators? If the latter, how many investigators documented cell adhesion events and were they blinded to the study hypothesis?

We have clarified these points in our revised manuscript: First, we have added further details of our static adhesion assay protocol to the revised Methods section and we have cited a methods chapter by Butler et al. published in *Angiogenesis protocols* (pages 231-248; https://doi.org/10.1007/978-1-4939-3628-1_16) that describes the protocol in more detail. Second, we have also more precisely defined the term “adhesion events” in the revised Results section. Briefly, adhesion events were counted manually by an unblinded investigator, and then confirmed by a blinded investigator, using the Image-Pro v6.3 software.

To demonstrate the validity of our results, we have included a table below that summarises the concordance of the adhesion event quantifications between the unblinded and blinded investigators for one of the biological replicates of the HCAEC static adhesion assays, which included 3 technical replicates per biological replicate. We found the quantifications of the adhesion events to be highly concordant between the investigators ($P > 0.05$, two-sided t-test).

Condition	Technical replicate	Scientist 1	Scientist 2	Difference	Replicate mean (sd)		P-value
					Scientist 1	Scientist 2	
Vehicle control	1	114.11	120.40	-6.30	121.45 (6.41)	125.39 (5.12)	0.2378
	2	125.91	125.12	0.79			
	3	124.34	130.63	-6.30			
TNF	1	391.90	399.77	-7.87	377.47 (22.32)	393.21 (10.03)	0.1569
	2	388.75	398.19	-9.44			
	3	351.76	381.67	-29.90			
APC + TNF	1	217.98	230.57	-12.59	234.77 (19.76)	238.97 (10.71)	0.5093
	2	256.54	251.03	5.51			
	3	229.79	235.29	-5.51			

11. Did the authors assess cell viability throughout these experiments?

Yes, we assessed cell viability in this study using Trypan Blue. We only performed our experiments in HUVECs and U937 cells with viability >95%. We also made sure to minimise the number of cell line passages to ensure maximal viability. HUVECs were discarded after passage 4, and U937 cells after passage 10. We have added these details to the revised Methods section of our manuscript.

12. The authors refer to other studies suggesting that endothelial cells exposed to APC have lower levels of expression of adhesion molecules such as ICAM-1 and VCAM-1. Could these molecules influence the apparently lower cell-adhesion observed in the presence of APC in this study?

We thank the reviewer for highlighting this interesting hypothesis. Indeed, as noted in the submitted version of the manuscript, previous studies have suggested that exposure of TNF-treated endothelial cell lines to APC attenuates the TNF-associated increase in mRNA expression and surface levels of cellular adhesion molecules such as ICAM-1 and VCAM-1 (Joyce et al. (2001) J Biol Chem. 276, 11199-11203). These findings had provided us with the motivation to investigate the effect of APC on leukocyte-endothelial cell adhesion.

To address the reviewer's comments, for our revised manuscript, we performed RT-qPCR to assess directly the effects of APC on *ICAM1* and *VCAM1*, as well as *CCL2* (encoding the monocyte chemoattractant protein 1) mRNA expression levels in TNF-treated HUVECs and HCAECs. As described in the revised Results section, in both TNF-treated HUVECs and HCAECs, we show that APC exposure led to a significant reduction in the mRNA levels of *ICAM1*, but not *VCAM1* or *CCL2*.

Together, these results indicate that ICAM-1 may be key in mediating the role of APC in leukocyte-endothelial cell adhesion.

13. The materials and methods section should be rearranged so that the methods for each part of the investigations is presented in a chronological order (i.e., begin with the phenome scan and finish by the functional experiments).

We thank the reviewer for this helpful suggestion and have rearranged the Methods section in the revised manuscript accordingly.

Reviewer #2 (Remarks to the Author):

1. The manuscript by Stacey et al. presents an investigation of an observation for the pleiotropic genetic variant rs867186 A>G (S219G) in the *PROCR* gene (endothelial protein C receptor, EPCR) which has a dual, opposing effects on coronary artery disease (CAD, lower risk) and venous thromboembolism (VTE, higher risk). The background and the rationale of the study are presented in a well-structured manner and the investigation of this cross-disease association is deemed to be of high importance.

We thank the reviewer for the positive feedback on our manuscript.

2. However, there are several major concerns with the experiments aiming to elucidate the mechanistic basis behind 219G association with lower CAD risk and higher VTE risk: As pointed out in the Introduction, the association of the 219G genotype with CAD and VTE is already known, and the analysis is essentially confirmatory.

We agree with the reviewer that the genetic associations of the *PROCR*-219G variant with CAD/VTE have been reported before, and we referred to these studies in the Introduction section. However, we have extended these analyses in several important dimensions to gain new insights: First, we performed a phenome-scan to assess the associations across 1,402 broad electronic health record-derived ICD-codes from the UK Biobank to show that the variant effects are specific to the circulatory system. Second, we reported associations with stroke phenotypes and contrast the findings to other cardiometabolic diseases. These data support the broader conclusion that individuals carrying rs867186-G alleles have lower susceptibility to arterial thrombotic diseases but a higher risk of venous diseases. Third, we performed a large-scale association analysis with intermediate traits related to the cardiovascular system, reporting that rs867186-G was not associated with conventional cardiovascular risk factors, including lipid levels, type 2 diabetes and hypertension, but with factors of the protein C pathway. Our analyses based on the largest dataset currently available allow for the interpretation and comparative assessment of the broad genetic effects of *PROCR*-219G. Forth, using statistical colocalization and Mendelian randomization analyses, we uncovered shared genetic aetiology across (activated) protein C, factor VII, CAD and VTE, identifying p.S219G as the likely causal variant at the locus and protein C levels as a causal factor in arterial and venous diseases. Finally, we conducted targeted experiments to gain new insights into the biological consequences of the *PROCR*-219G variant.

3. The most interesting aspect is the analysis of the molecular mechanism behind these seemingly opposing effects. Unfortunately, the data to support any sort of mechanism is only superficial and definitely not conclusive.

In response to the reviewer's comment, we have conducted further experiments to elucidate the molecular mechanism underlying *PROCR*-rs867186 (as outlined below), and rephrased the text to acknowledge that our findings require further independent validation and follow-up experiments.

Other major issues:**PC/APC, EPCR/sEPCR levels and coagulation**

4. Figure 2C demonstrates that protein C (PC) is elevated in the 219G genotype shown by all ELISA, SOMAScan and chromogenic assay. What is the mechanistic explanation for that? How will the EPCR missense variant S219G lead to increased PC levels? Why are the levels of PC not measured by an immunoassay in the recall-by-genotype study? The finding that PC levels/activation is increased in the

219G genotype do not match the notion that this allele is associated with higher thrombosis (i.e. VTE) as the opposite will be expected.

Our complementary analyses using data derived from three different protein assays revealed a consistent effect of the *PROCR*-219G variant on soluble protein C (PC) levels and activity. Mechanistic studies demonstrated that the variant results in increased shedding of EPCR from the endothelial surface by rendering the receptor more sensitive to cleavage by metalloprotease (Qu et al. (2006) *J Thromb Haemost.* 4, 229-235) and by forming an alternatively spliced, truncated transcript (Saposnik et al. (2008) *Blood* 111, 3442-3451). The reduced levels of membrane-bound EPCR levels lead to an accumulation of PC because only the membrane-bound form of EPCR is able to convert PC into APC (Stearns-Kurosawa et al. (1996) *Proc Natl Acad Sci (USA)* 93, 10212-10216). We have now added this mechanistic explanation to our revised Discussion section.

Previous studies investigating the association between rs867186 and PC levels utilised either an immunoassay or a chromogenic assay. In our view, both are established and well-standardised methods. We opted to use the chromogenic assay (1) to rule out potential epitope effects and (2) to leverage in-house expertise with the assay at the Specialist Haemostasis Unit at the Cambridge University Hospitals NHS Foundation Trust. The chromogenic assay enables quantification of PC levels by a two-step process. First, PC in platelet-poor plasma is exogenously activated, and then the concentration of PC is determined by the colour change from a chromogenic substrate for activated PC using a standard reference curve. In this way, the chromogenic assay provides a measure of the levels of PC in a plasma sample with activation potential.

We have discussed the finding that *PROCR*-219G is associated with higher PC levels/activation as well as increased risk of thrombosis below.

5. The findings about activated PC (APC) levels in 219G genotype are discrepant: APC was shown to be upregulated by SOMAScan but not by immunoassay (Figure 2C and Figure 5). 219G genotype exhibits an elevated soluble EPCR (sEPCR) and even a “truncated” EPCR according to the Introduction – this “dysfunctional” EPCR will be expected to decrease APC formation, why is it increased or unchanged instead? APC is an anticoagulant – why its increase will be associated with higher thrombotic potential?

We agree with the reviewer that the SomaScan and immunoassay findings for APC provided in the manuscript appear to be discrepant. A likely explanation for this observation is the difference in statistical power for the analyses of the data generated with the SomaScan and immunoassays, performed in 3,301 and 52 individuals, respectively. We performed a post-hoc power calculation for the recall-by-genotype study, see below, and provided further context and limitations of our analysis in the Discussion section.

To confirm that the data from the SomaScan assay were sufficiently powered to detect the reported genotypic effect on APC, we performed a *post-hoc* power calculation using the following parameters: (i) $\alpha=0.05$, (ii) observed $\beta=0.256$, (iii) $n=3,301$, (iv) rs867186 minor allele frequency (MAF)=0.1. This calculation showed that we had 100% power to detect the observed association between rs867186 and APC. Furthermore, a sample size of 3,301 individuals afforded us 80% power to detect a genotypic effect size of $\beta=0.032$, which is almost an order of magnitude smaller than the observed effect size ($\beta=0.256$).

Post-hoc power calculations for the recall-by-genotype study using effect sizes obtained from previous studies. We compared the statistical power in a cohort consisting of an equal number of major and minor homozygotes of a genetic variant of interest. We used the absolute per-allele effect sizes (β) for sEPCR, PC activity and APC from the recall-study. The effect size of FVIIa is provided by the CHARGE consortium (for details see Suppl. Table 1). The calculations are independent of the allele frequency of the variant. The x-axis is the total sample size of the recall experiment; the number of eventually recruited homozygotes ($n=36$) is indicated with a dashed line. The y-axis is the statistical power; 80% is indicated with a dashed line. We assume a type I error rate of $\alpha=0.05$. The power calculation is for an equal variance two-sample two-tailed t-test, as further described in Corbin et al. (2018) *Nature Commun.* 9, 711.

We agree that the direction of the effect observed in the SomaScan data appears to be counterintuitive. To this end, we had performed extensive validation experiments to assess the binding specificity of the APC SOMAmer. The data confirmed that the SOMAmer binds APC with high specificity (Methods and Suppl. Table 2).

While we agree that the primary effect of membrane EPCR dysfunction would be reduced APC levels in an endothelial cell model, we cannot discount the possibility of secondary or compensatory effects that would likely occur *in vivo* and may impact APC levels. We note that the primary driver of PC activation is the thrombin-thrombomodulin complex. Indeed, the thrombin-thrombomodulin complex increases the rate of PC activation by ~1000-fold, whilst the involvement of EPCR further enhances activation rates by ~10-fold. In our revised Discussion section we have added some suggestions as to how APC may be increased in carriers of *PROCR-219G*. Although beyond the scope of our study, further work is needed to replicate this SOMAscan finding in the first instance, and to then elucidate the precise mechanisms using an *in vivo* model.

- "To confirm the specificity of the binding events, we measured the binding activity of the PC and APC SOMAmer reagents to a range of relevant proteins, specifically, PC, APC, sEPCR, thrombin, FV, FVIIa, Protein S and thrombomodulin". None of these other protein associations were independently confirmed by immunoassays. Given the observed discrepancy of the SOMAmer measurements with immunoassay for activated protein C, this should be done.

In the submitted version of the manuscript, we assessed the possible effects of a range of proteins on the binding activity of the PC and APC SomaScan signals. Only the PC and APC SOMAmers were assessed, because only these two SOMAmers showed a genetic association signal in our analysis shown in Fig. 2C ($P=0.05/30 \text{ traits}=1.67\times 10^{-3}$).

7. The SOMAmer assay provides measurements of hundreds or thousands of proteins. The associations of the minor (G) allele of rs867186 (*PROCR*-219Gly) with all measurements should be presented.

We thank the reviewer for this suggestion. We have added Suppl. Table 5 to the revised manuscript that provides an overview of the associations ($P<0.05$) of *PROCR*-rs867186 with protein measurements of the SomaScan platform (based on two independent cohorts, INTERVAL [Sun et al. (2018) *Nature* 558(7708), 73-9] and KORA [Suhre et al. (2017) *Nat Commun.* 8, 14357]). Note that the associations with PC and APC are the only pQTLs that pass the genome-wide significant threshold ($P<5\times 10^{-8}$).

8. Previous studies have shown that physiologically high sEPCR does not affect the anticoagulation potential of the protein C pathway as only the membrane EPCR (mEPCR) is important for APC formation and anticoagulation. An increase of APC will therefore be due to higher mEPCR which together with the higher sEPCR will be indicative of higher total levels of EPCR in 219G genotype. Authors should specifically investigate the mEPCR in their in vitro studies – is the EPCR expression changed in 219G genotype? Are the mEPCR levels changed in 219G genotype? Are the sEPCR levels elevated at the expense of a reduced mEPCR? Authors should remove the diagram showing reduction in mEPCR on Figure 7 since this has not been demonstrated in the study (or otherwise investigate it).

In the submitted version of the manuscript, we showed that *PROCR*-219G genotype is associated with higher levels of sEPCR (Fig. 2A and 5A). These findings are concordant with previous data (e.g. Kallel et al. (2012) *BMC Med Genet.* 13, 103).

In the revised manuscript, we have now performed additional experiments to confirm that EPCR expression is reduced in HUVECs derived from donors who are homozygous for *PROCR*-219G (n=3) compared to donors who are homozygous for *PROCR*-219S (n=3) (Methods). In flow cytometric analysis, we found a 1.9-fold reduction in mEPCR levels in homozygotes of 219G relative to homozygotes of 219S under basal conditions, as well as a 1.8-fold reduction after cells were treated with phorbol 12-myristate 13-acetate (PMA), a potent agent to enhance ectodomain shedding. These data suggest that *PROCR*-219G promotes cellular shedding of EPCR, and is not associated with higher total levels of EPCR. We have added these data to the revised manuscript as Figs. 4B and C.

To further address the reviewer's assertion that higher levels of APC would be consistent with higher total levels of EPCR in carriers of *PROCR*-219G, we obtained gene expression QTL data from human aortic endothelial cells (HAEC) in 147 donors (Erbilgin et al. (2013) *J Lipid Res.* 54, 1894-1905). We observed a significant reduction in *PROCR* mRNA levels in 219G carriers, suggesting there are lower total EPCR levels in 219G carriers. This observation was consistent in basal and proatherogenic (i.e. stimulated with oxPAPC for 4 hr) HAECs, with $P=1.25\times 10^{-21}$ and $P=9.71\times 10^{-21}$ (non-parametric linear model), respectively.

Furthermore, as noted above, PC activation is primarily driven by the thrombin-thrombomodulin complex, so an increase in APC levels is not necessarily dependent on increased mEPCR. Therefore, secondary or compensatory mechanisms may be responsible for the increase in APC levels.

9. EPCR can also be expressed on platelets (especially activated platelets). Blood processing in this study does not ensure that plasma is free of platelets: firstly, spinning blood at 4 deg would likely lead to platelet activation, and secondly, 1,000g for 15 min will not provide a platelet-free plasma. What is the evidence that EPCR measured in plasma is soluble and not platelet-associated?

We thank the reviewer for highlighting this point. We agree that the blood processing in our study likely resulted in platelet-poor rather than platelet-free plasma. However, the collection of blood samples in sodium citrate tubes ensured limited *ex vivo* platelet activation. In our view, the presence of EPCR expressed on activated platelets in our plasma samples would have merely led to a dilution of the genotypic effect of rs867186 on sEPCR levels, rather than to a false positive result, as the presence of platelets would have likely affected the EPCR expression levels irrespective of 219G carrier status. Therefore, we are confident that our conclusions concerning the genotypic effect of 219G on sEPCR plasma levels are valid.

10. FVII is also elevated in the 219G genotype. What is the mechanistic explanation for that? How is this affecting protein C pathway? For instance, FVII is a known ligand for EPCR where it can displace PC and decrease its activation, thus leading to a procoagulant effect. Does the missense S219G variant affect FVII binding to EPCR? FVII and its interactions with EPCR have not been studied at all while there is a very vague argument about its involvement in the Discussion which is hardly supported by any data.

In the Discussion section of the submitted manuscript, we proposed the mechanistic explanation for the higher FVII levels to be due to reduced endocytosis of EPCR-FVII complexes in carriers of the 219G genotype (see also reply below). Indeed, this mechanism would also be consistent with the higher PC and APC levels that we detected in plasma samples collected from this genotypic group. As the reviewer noted, previous studies showed that FVII is a ligand for EPCR. We have now expanded the Discussion section in the revised manuscript to more broadly report the evidence from literature, and suggested further experiments that could be conducted to prove conclusively this experimental hypothesis, e.g. through performing endocytosis assays in genotype-specific or CRISPR/Cas9-edited endothelial cell lines.

11. Authors mention that FVII-EPCR interaction leads to clearance by endocytosis. Hence, 219G genotype should have more EPCR endocytosis due to higher FVII being present. At the same time authors show higher sEPCR in 219G – these are two contradicting arguments.

In our revised manuscript, we have now provided experimental data showing a reduction of membrane-bound EPCR levels on HUVECs obtained from individuals homozygous for *PROCR*-219G. These data are consistent with previous reports (e.g. Medina et al. (2014) *Arterioscler Thromb Vasc Biol.* 34, 684-690). We have summarised our findings in the revised Results section and Figs. 4B and C.

Since membrane-bound EPCR is endocytosed when complexed with FVII (or PC) (Nayak et al. (2009) *Blood* 114, 1974-1986), reduced levels of membrane-bound EPCR would be expected to result in reduced endocytosis of EPCR-FVII complexes. This is consistent with higher circulating levels of FVII.

12. There is also a speculation about the increased D-dimer in the Discussion which is similarly vague and unproven. Authors should remove the diagram showing fibrin deposition increase in 219G genotype on Figure 7 unless they specifically investigate this (e.g. fibrin formation assay).

D-dimer is a fibrin degradation product and a widely used clinical marker of activated blood coagulation. The paragraph in the Discussion section referring to the association of higher plasma levels of D-dimer in rs867186-G-allele carriers was based on evidence from a previous epidemiological study that specifically investigated this association (Smith et al. (2011) *Circulation* 123, 1864-1872). In this meta-analysis of GWAS with a total of 21,207 individuals, the rs867186-G showed a per-allele effect of 0.05 log(ng/dl) ($P=3.70 \times 10^{-6}$). In the contributing cohorts, D-dimer levels were measured by ELISA using monoclonal antibodies. In the Discussion section of the revised manuscript, we have clarified that the association with D-dimer was indeed based on observational data, and we have also revised Fig. 6 with annotations indicating the type of evidence (e.g. literature or findings from our study) supporting each molecule included.

13. An important thing which is missing is a coagulation assay to show whether plasma from 219G genotype indeed has higher propensity to form clots.

We share the reviewer's view that data supporting the notion that the *PROCR*-219G genotype has higher propensity to form clots is important. To this end, in the revised manuscript, we now also report the association with prothrombin time (by international normalized ratio, INR), which measures the clotting time in plasma after the addition of tissue factor and reflects the integrity of the extrinsic and common coagulation pathways. In a GWAS of a community-based cohort of 1,218 individuals, rs867186-G was associated with shorter prothrombin time by INR ($\beta=-0.020$, $P=9.98 \times 10^{-8}$) (Tang et al. (2012) *Am J Hum Genet* 91, 152-162). In the Discussion section, we also cite a GWAS in a Japanese sample that found a genome-wide significant association ($P=6 \times 10^{-24}$) between rs867186 and prothrombin time with the same directionality (Kanai et al. (2018) *Nature Genet.* 50, 390-400). These observations are consistent with the finding that rs867186-G was also associated with higher levels of FVII antigen and higher FVII activity (Fig. 2C), as the prothrombin time is affected by the functional levels of FVII. These data, together with the observation of higher D-dimer levels, provide evidence that carriers of *PROCR*-219G have an increased clotting tendency of blood.

Flow cytometry

14. EPCR expression is 75% higher in HUVECs compared to monocytes – how is this calculated? The use of the blocking antibody demonstrates specificity but its use to estimate the level of expression on different

cell types is difficult to understand. Despite the fact that primary monocytes/neutrophils show lower signal for EPCR on Figure S6, practically all of the cells are positive for EPCR. Authors are encouraged to investigate the difference in EPCR expression using an alternative method as well.

The percentage of ~75% higher EPCR expression was calculated by comparing the median fluorescence intensity (MFI) for untreated HUVECs in Suppl. Fig. 5 (MFI: 21,117.47) with the MFI for untreated U937 cells in Suppl. Fig. 6 (MFI: 4,929.84). We acknowledge that this statement was unclear; therefore, we have removed it from the revised text.

With regards to our use of blocking antibodies, we can confirm that we used them solely to ascertain the specificity of antibody binding within cell types and not to estimate surface levels across different cell types.

Finally, we do indeed detect a signal from two APC-conjugated EPCR antibodies (derived from distinct clones) in both primary human monocytes and neutrophils. However, when we assessed the specificity of this signal using unconjugated blocking antibodies, we did not see the expected downward shifts in signal with either conjugated antibody (Suppl. Fig. 7). Conversely, when compared with Suppl. Figs. 4 and 6, the blocking antibody did induce the expected downward shift in signal in both HUVECs and U937 cells. This indicates that the signals from these antibodies in HUVECs and U937 cells are specific to EPCR, whereas in primary cells, the small signal is due to non-specific binding. To investigate this further, we performed various follow-up experiments. As shown below, we observed a small signal from isotype control antibodies in primary human monocytes and neutrophils of a similar magnitude to the EPCR signal, whilst the addition of an Fc receptor blocking antibody control did not affect the signal from either the EPCR or the isotype control antibody. Notably, the MFIs (X-Med) did not change with the use of an Fc receptor blocking antibody control.

Monocytes:

Neutrophils:

- Monocytes express Fc receptors and therefore incorporation of an Fc receptor blocking antibody control is desirable to show specificity of the signal.

We addressed these points in our reply above.

Inflammation/CAD

- The leukocyte-endothelial cell adhesion assay is poorly designed, and the conclusions are purely speculative. Firstly, this is a static assay performed by addition of macrophages on top of a monolayer of HUVECs. However, authors state that their assay is performed as previously described in two articles (ref 77 and 78) which conduct an adhesion assay under flow. Rolling, arresting and transmigration cannot be

examined using a static assay as authors say they did. Rolling and arresting occur under flow conditions, while transmigration requires the cells to extravasate in another basal compartment, which is not possible when the HUVECs are seeded in a 2D tissue culture dish. What was quantified exactly in this model and how? Cells are used at $\geq 90\%$ confluency so how do authors ensure macrophages do not bind to the 10% of the tissue culture plastic? Furthermore, washing the cells “gently” with saline is likely to be highly inconsistent. Usually static assays are performed by inverting a plate and centrifuging it to remove non-adherent cells or similar. A flow assay should be performed to confirm that there is indeed an effect of APC on adhesion or transmigration of macrophages.

We thank the reviewer for these very helpful comments. In our revised manuscript, we have now removed the two references mentioned above, as these do indeed refer to the flow assay method. Instead, we have referenced a methods chapter by Butler et al. published in *Angiogenesis protocols* (pages 231-248; https://doi.org/10.1007/978-1-4939-3628-1_16) that describes the protocol for static adhesion assays (see Section 3.3 of Butler et al.) that we followed in our study. This includes gentle aspiration of PBSA during rinsing steps. We did consider inverting the plates to perform rinsing steps, as the reviewer suggested, but we felt this was only suitable in a 96-well plate set-up. We used 12-well plates in our experiments.

It is noteworthy that in earlier optimization experiments we ran some flow adhesion assays using both PBMC-derived monocytes and neutrophils, as well as PMA-differentiated U937 cells as per our static adhesion assays. However, in contrast to the PBMC-derived cells, stimulated U937 cells did not exhibit any transmigration and almost no rolling. Since we aimed to utilise the U937 cell line to avoid potential confounding genotypic effects from the use of PBMC-derived cells from different donors, we decided to utilise the static adhesion assay design to assess ‘adherence’/‘adhesion’ only. Adhesion events were counted manually using Image-Pro v6.3 software, which provides measurement tools designed specifically for counting cellular events such as adhesion. Further, in our revised submission, we have now removed any reference to ‘rolling, arresting and transmigration’, in favour of the term ‘adhesion’.

Finally, we did not observe any U937 cells adhering to the plastic, but since the confluence of the HUVEC/HCAEC monolayers were highly comparable across conditions in each experiment, any U937 cells adhering to the plastic should be consistent across conditions.

17. HUVECs are not a suitable model for the research question of this study – these are foetal venous cells which are not related to atheromatous formation in coronary arteries. Arterial cells should be used, e.g. coronary artery endothelial cells, in order to provide robust evidence for effects in CAD. This is especially important for this study where CAD and VTE are compared.

We thank the reviewer for this suggestion. As recommended, in the revised manuscript, we have now reported on RT-qPCR experiments and static adhesion assays using both HUVECs and human coronary artery endothelial cells (HCAECs) (Fig. 5). Note that the results observed in both cell models were very similar, providing additional support to the mechanistic hypothesis.

18. Which surface adhesion molecules are reduced by APC treatment? Authors should show this by e.g. qPCR and western blotting. Since APC is only present during the cell co-incubation and the adhesion assay only lasts 5 min, how is it possible for APC to induce any transcriptional or translational changes?

We thank the reviewer for this suggestion. In the revised manuscript, we have now performed RT-qPCR to quantify *ICAM1*, *VCAM1* and *CCL2* gene expression in both HUVECs and HCAECs following co-incubation with TNF and APC for 24 hr. We showed that APC significantly attenuated the effect of TNF

on *ICAM1* mRNA levels (in both HUVECs and HCAECs) in a dose-dependent manner, but not *VCAM1* or *CCL2* mRNA levels (Fig. 5A). The data are presented in the Results section entitled “Effect of APC on cell adhesion molecule expression and leukocyte-endothelial cell adhesion”. Consistent with previous data (e.g. Joyce et al. (2001) J Biol Chem. 276, 11199-11203), our data strengthen the evidence that ICAM-1 is a key mediator of the effects of APC on leukocyte-endothelial cell adhesion.

We note that in the adhesion assays involving APC exposure reported in our revised submission, HUVECs were actually co-treated with APC and TNF for 24 hr, which was followed immediately by the adhesion assay. This prolonged APC treatment was to allow time for transcriptional effects to occur. The experimental protocol and rationale have been made clearer in the revised manuscript.

19. A single finding of an in vitro adhesion assay which was not designed well cannot be generalised to indicate effects on atherosclerosis, a complex multifactorial disease involving multiple cell types and risk factors. Such conclusions can only be made if multiple models are used – e.g. use of diseased cells, use of cells with 219G genotype, use of cells overexpressing EPCR or with downregulation of EPCR, murine model of atherosclerosis to show that 219G indeed leads to increased atheromatous lesion formation etc.

We fully agree with the reviewer that additional *in vitro* and *in vivo* experiments are required to provide further insights into the molecular mechanism underlying *PROCR-219G*. We have followed the reviewers’ suggestion and extended the scope of the adhesion assay to include additional replicates for HUVECs and to validate the findings in another cellular model, HCAECs. We also extended the phenome-scan and Mendelian randomization analyses. Taken together, we believe we have provided sufficient data to support a plausible molecular mechanism underlying *PROCR-219G*. We have made it clear in the Discussion section of the revised manuscript that more work is needed to elucidate further molecular, cellular and physiological aspects of this proposed mechanism.

20. The authors’ conclusions about the findings of the in vitro adhesion assay are an overinterpretation of the data. There is no anti-inflammatory role of APC shown in this manuscript and this statement should be removed. The hypothetical activation of PAR-1 by APC and reduction of ICAM-1 and VCAM-1 explained in the Discussion is speculative and practically impossible to occur in the adhesion assay – APC cannot have transcriptional effects on the endothelial cells in a 5-min assay. The speculation about APC-PAR-1 interaction, the statement that APC has a protective role in atherosclerosis and the diagram showing PAR-1 signalling and reduction of vascular inflammation on Figure 7 should all be removed or specifically investigated prior to inclusion.

We acknowledge that the involvement of PAR-1 signalling and the subsequent reduction in cellular adhesion molecules such as ICAM-1 and VCAM-1 during this cellular cascade were based on previous studies. However, as noted above, we have provided experimental evidence that APC does indeed exert a significant effect on *ICAM1* mRNA levels in TNF-treated HUVECs and HCAECs. Importantly, in these experiments, we treated cells with both APC and TNF for 24 hr prior to cell lysis, providing an appropriate timespan for transcriptional events to occur. Furthermore, our statement that APC has a protective role in atherosclerosis is supported by our Mendelian randomization analyses (Suppl. Table 4). Nevertheless, we have now revised Fig. 6 (see below) to make it clearer that our conclusions are supported by our findings and/or findings from previous studies.

Figure 6. Proposed molecular mechanism underlying the PROCRR-p.S219G variant. *Credits: The images in this figure were retrieve from Flaticon (<https://flaticon.com>).*

21. The statement at the bottom of page 9 that 219G genotype has lower susceptibility to atherosclerosis due to increased APC levels contradicts the findings on Figure 5 which show no APC elevation in 219G genotype.

The finding that carriers of the PROCRR-219G genotype have higher APC levels is supported by the SomaScan data, presented in Fig. 2C. We have rephrased the sentence accordingly. The discrepancy between the findings from the SomaScan and chromogenic assays are discussed above.

Minor issues:

22. Methods state that both plasma and serum have been collected. Which are the experiments performed with serum samples? What was the rationale behind this?

We collected serum samples from all participants but have not used these samples for the work presented here. The samples were collected for potential further studies. In the revised manuscript, we have removed the statement that we collected serum samples from the Methods section.

23. Please correct cell seeding densities – cell density should be expressed as cell/cm² for adhesion cells or cells/ml for cells in suspension.

We thank the reviewer for pointing this out. As advised, we have made the relevant change to seeding densities in the revised text.

24. Please use the same scale for X-axes on Figures S4, S5 and S6 and make the graphs equal in size.

As noted in the Methods section in the submitted version of the manuscript, we performed flow cytometric analyses using two different instruments. This was because our Cytomics FC500 instrument was decommissioned midway through our study; hence, all outstanding flow cytometry experiments were performed using a CytoFLEX S. Specifically, Suppl. Figs. 5 and 6 were prepared using data acquired on the CytoFLEX S. This is the reason for the slightly different X-axis scales and labels for Suppl. Figs. 4 and 7 vs. 5 and 6. We have now altered the scales manually in Kaluza, and as suggested, we have also made the graphs equal in size in the revised Supplementary Information.

25. Please state in the Methods section the concentration of sEPCR, APC and Mac-1 used in the adhesion assay. Please include details of the timing of addition of sEPCR, APC and Mac-1.

We have made the relevant changes in text.

Reviewer #3 (Remarks to the Author):

1. The authors eloquently present the results from a compelling series of experiments that dissect the pleiotropic and seemingly paradoxical effects of the *PROCR*-p.S129G variant on CAD and VTE. They use a combination of approaches from molecular epidemiology, statistical genetics, recall-by-genotype deep phenotyping, and cell culture assays to tease apart the mechanisms underlying the pleiotropy at this variant. The statistical analyses are well conceptualized, appropriately conducted and reported, and clearly presented, as is the translational and wet-bench science. The work critically advances our understanding of the locus and identifies the need for more detailed mechanistic studies. Nonetheless, I do have the following critiques:

Is sEPCR targeted by the SOMAscan platform and if so, are levels associated with rs867186 in data from KORA and/or INTERVAL and does the genetic signal in that region colocalize with CAD, VTE, PC, and APC?

We thank the reviewer for this suggestion. Unfortunately, there is no corresponding sEPCR SOMAmer reagent on the SomaScan platform that could be used for our analyses. Likewise, sEPCR is not targeted on the Olink protein panels either. We hope that the protein will be included in future versions of the proteomics assays to investigate this question.

2. The authors clearly demonstrate the statistical evidence of the rs867186 variant being causal for PC, APC, and FVIII levels, and DVT and CAD outcomes, as well as nice statistical evidence of PC and APC levels being causal for CAD and VTE. In the later experiment they used a series of independent variants in *PROCR* to instrument the exposure. Is it possible to further refine these analyses to more directly demonstrate a causal pathway from rs867186 to PC/APC to outcomes using a mediation-based approach?

We thank the reviewer for suggesting to perform a mediation analysis. We fully agree that this is the most informative statistical approach to ascertain whether PC/APC lies on the causal pathway toward CAD/VTE. This analysis requires the concomitant availability of PC level and CVD outcome data in the same individuals. To this end, we have conducted the suggested analysis in participants of European ancestry of the Atherosclerosis Risk in Communities (ARIC) Study. Specifically, the following formula was used, where the *PROCR* genetic score was the same as used for the MR analyses:

$$\text{coxph (CHD or VTE} \sim \text{PC1} + \text{PC2} + \textit{PROCR} \text{ genetic score} + \text{sex} + \text{age})$$

$$\text{coxph (CHD or VTE} \sim \text{PC1} + \text{PC2} + \textit{PROCR} \text{ genetic score} + \text{protein C} + \text{sex} + \text{age})$$

However, we found that the unadjusted genetic association of the *PROCR* genetic score with CHD or VTE outcomes was not statistically significant ($P > 0.05$). This is likely due to (1) the relatively low number of CHD and VTE patients in the ARIC study for which protein C levels were also available, i.e. $n=1,295$ and $n=528$, respectively; and (2) the relatively large within-individual variation of the protein C measurements. Therefore, we were unable to determine whether the associations with CAD/VTE are attenuated on adjustment for protein C levels. We hope that it will be possible to conduct a mediation-based approach in future studies when larger sample numbers are available.

3. The authors performed MR for PC and APC effects on cardiovascular outcomes and then reverse MR to exclude bidirectional effects. It seems odd that Table 1 has the Forward MR for PC and the Reverse MR for APC. In the absence of a compelling reason for this it might be more appropriate to have the table reflect either PC or APC and then have the other species in the supplement.

We thank the reviewer for this suggestion. We have now retrieved the full GWAS summary statistics for Protein C (Tang et al (2010), Blood 116, 5032-5036), which allowed us to perform the reverse MR analyses. As suggested, in the revised manuscript, we present both the forward and reverse MR results for PC in Table 1. The forward and reverse MR results for APC are shown in Suppl. Table 4.

4. The authors are to be commended for the inclusion of recall-by-genotype analyses and I am in full support of the approach they present in Figure 1; it is great to see it in practice. Their discussion of the generalizability of the approach they use in the paper is spot on and well taken. The manuscript would benefit, however, from additional context about what the authors feel recall-by-genotype analyses added to this specific study beyond what is already known or otherwise presented in the manuscript. As the authors explain in the introduction, the PROCR-p.S129G has already been well characterized and is known to result in increased sEPCR. The increased PC activity, which is essentially a proxy for PC levels, is clearly shown earlier in the paper. And they do not detect an effect of genotype on the TAT complex or APC in the recall-by-genotype analyses.

We are grateful to the reviewer for the supportive comments on our study design. The rationale for conducting the recall-study was (1) to replicate the previous findings under strict control of experimental conditions permitted by the recall-study, e.g. PC and APC; and (2) to expand on the panel of biomarkers relating to the protein C pathway that cannot be readily assessed through large-scale proteomics platforms, including sEPCR and TAT complex.

Our recall-study did indeed provide replication of the genotypic association with PC in an independent sample. Furthermore, we used a chromogenic assay to measure PC activity (as a marker of PC levels), which is not affected by potential binding-affinity effects of protein-altering variant (such as rs867186) often detected in protein-binding assays (such as SomaScan assay and immunoassays, as presented in the phenome-scan). We have added this detail to the recall-study section of our revised manuscript.

5. How do the authors explain the apparent discrepancy between the epidemiological analyses and recall-by-genotype based analyses with respect to the rs867186 – APC relationship?

The SomaScan data were derived from a much larger sample relative to our recall study, i.e. 3,301 vs. 52 individuals, respectively. Therefore, the SomaScan data were better powered to detect a genotypic effect, which likely explains the discrepant findings we observed in our study. We have added the power calculation on the revised version of the manuscript and added in the study limitations to the Discussion section. We would also direct the reviewer to a reply to a similar comment from reviewer #2 above.

6. The putative mechanistic pathway that the authors propose for PROCR to increase VTE risk seems to be centered on increased sEPCR resulting in decreased FVII, likely through decreased membrane EPCR. The evidence here seems to come from pre-existing literature, the recall-by-genotype studies showing increased sEPCR in rs867186 carriers, and the epidemiological association of rs867186 with decreased FVII levels. Is there evidence either in the literature, or that the authors can provide experimentally, that demonstrates that rs867186 is, in fact, associated with decreased surface levels of EPCR and thus link the

increased levels of sEPCR with decreased FVII mechanistically, rather than just hypothesize that increased sEPCR results in decreased membrane EPCR?

We thank the reviewer for highlighting this important point. In the revised manuscript, we have now conducted experiments utilising rs867186-genotype-specific HUVECs to show that *PROCR*-219G is associated with lower membrane EPCR levels. We have summarised these findings in the Results section entitled “Quantification of EPCR expression and shedding in endothelial cells” and Figs. 4B and C. Our results are consistent with previous *in vitro* studies (Medina et al. (2014) *Arterioscler Thromb Vasc Biol.* 34, 684-690).

Further, we have expanded the Discussion section to elucidate the mechanistic connection between lower mEPCR and higher circulating FVII levels. Note that our phenome-scan indicated that FVII levels and activity were *higher* in *PROCR*-219G carriers.

7. Additionally, the pathway the authors posit for rs867186 leading to VTE does not seem to take into account the MR experiments suggesting that PC and APC are causal for these outcomes. How do they reconcile the results of the MR experiments with their proposed mechanism which appears to be independent of actual PC/APC levels?

Our analyses provide evidence of causal relationships between the levels of zymogenic and activated protein C and CAD and VTE outcomes, in opposite directions. We agree that further work is needed to elucidate the molecular mechanism of how PC/APC levels may directly influence VTE risk. To this end, we have now expanded the Discussion section to provide a mechanistic hypothesis that can be addressed in future experiments.

8. The manuscript would be improved by presenting the putative mechanistic chain with a bit more circumspection. There are many steps in Figure 7, especially with respect to VTE, that are not clearly demonstrated in the paper and remain hypotheses. These steps in the mechanistic chain will likely require experiments using genetically edited iPSC, genotype specific cellular assays, or model systems to test. Understandably these experiments are most likely beyond the scope of this paper but I think more explicitly recognizing the remaining uncertainty, and where in the pathway it lies, would make the paper stronger.

We agree with the reviewer, and have performed additional experiments to improve our understanding of the proposed molecular mechanism underlying *PROCR*-rs867186 (see details above). Further, we have prepared an improved version of Fig. 6 that more clearly indicates which data were derived from (1) our *in vitro* experiments, (2) genetic association data, (3) previous studies in the literature, or (4) represent hypotheses that necessitate further experiments.

Reviewers' Comments:

Reviewer #1:

None

Reviewer #2:

Remarks to the Author:

The authors have carefully revised the manuscript.

1) The addition of in vitro data presented in Figure 4 and 5 was necessary to improve the manuscript but the present experiments are clearly underpowered (currently only n=2-3 per group). I would recommend a higher number of biological replicates (at least n=5 per group).

2) Also, results should be analysed with a two-tailed, rather than a one-tailed t-test. The authors also state that these groups were tested for normal distribution using the Shapiro-Wilk test. I would have thought that performing this test on such a small sample size is futile?

3) Response to Reviewer 2 Comment 5: "We agree with the reviewer that the SomaScan and immunoassay findings for APC provided in the manuscript appear to be discrepant. A likely explanation for this observation is the difference in statistical power for the analyses of the data generated with the SomaScan and immunoassays, performed in 3,301 and 52 individuals, respectively."

Directionality should not be different because of lack of statistical power due to differences in sample sizes. The Somascan analysis might have more samples but if the measurements cannot be validated with an immunoassay, then the aptamer results are questionable and may not reflect APC levels.

4) The response to the platelet contamination issue is also not very convincing, and the authors should discuss this in the limitation section.

Reviewer #3:

Remarks to the Author:

The authors have addressed all of my concerns. The manuscript is much improved.

Elucidating mechanisms of genetic cross-disease associations: an integrative approach implicates protein C as a causal pathway in arterial and venous diseases

Rebuttal letter #2

Reviewer #2 (Remarks to the Author):

1. The authors have carefully revised the manuscript.

The addition of *in vitro* data presented in Figure 4 and 5 was necessary to improve the manuscript but the present experiments are clearly underpowered (currently only $n=2-3$ per group). I would recommend a higher number of biological replicates (at least $n=5$ per group).

We thank the reviewer for acknowledging that we have carefully revised the manuscript and that it has improved.

Below, we have addressed the reviewer's comment that "the present experiments are clearly underpowered" for each experiment presented in Figures 4 and 5 separately.

Recall-by-genotype experiments (Figure 4a). We measured four plasma biomarkers related to the functional state of the protein C pathway in 52 individuals. In the previous rebuttal letter, we have commented on the statistical considerations for the recall-study and performed a power calculation. We also noted in the Discussion section that "replication in independent large cohorts is needed".

EPCR expression in genotype-specific HUVECs (Figures 4b and 4c). For the experiments shown in Figures 4b and 4c, we used 3 biological replicates per genotypic group (i.e. distinct donors) and performed 3 technical replicates for each biological replicate to reduce technical variation. The experiments showed a statistically significant effect of rs867186 on EPCR expression ($P<0.05$). As discussed in our manuscript, the observed effect in our experiments was directionally consistent with the findings from two previous *in vitro* studies (Qu et al. (2006), *J Thromb Haemost.* 4, 229-235 and Medina et al. (2014), *Arterioscler Thromb Vasc Biol.* 34, 684-690). In summary, the reported finding has now been replicated across 3 independent studies. Therefore, we believe that our conclusions, which are based on robust and highly reproducible observations, are supported by the presented data.

qPCR experiments (Figure 5a). As noted by the reviewer, we ran originally 2–3 biological replicates for the qPCR experiments. We have now strengthened our findings by adding another 2 biological replicates, yielding 4–5 biological replicates in total. Importantly, we performed these additional biological replicates in two cellular models, i.e. HUVECs and HCAECs. The observed effect on *ICAM1* mRNA levels was replicated across both cellular models. We also note that we applied a gradient design, treating endothelial cells with 5 different APC concentrations (i.e. 0, 0.1, 1, 10 and 100 nM). This design provided additional, internal replicates as well as important data on the dose-response relationship.

Adhesion assays (Figure 5b). As for the qPCR experiments, we have strengthened the findings from our adhesion assays by adding additional biological replicates, with a total of 4–5 biological replicates per experiment. We also performed these additional replicates across both HUVECs and HCAECs, with the cellular effect again replicating in both cellular models. With these additional replicates, we are now able to present the statistical significance of this effect across both endothelial cellular models. In contrast to the qPCR experiments, here, we opted to run technical replicates (between 2–4 for each biological

replicate) of a single APC concentration (100 nM) rather than a gradient design due to the time-dependent, labour-intensive and serial nature of the adhesion assays.

Taken together, we have delivered the additional biological replicates, as recommended by the reviewer. We believe that we now present data from sufficient biological and technical replicates to draw unequivocal conclusions.

2. Also, results should be analysed with a two-tailed, rather than a one-tailed t-test. The authors also state that these groups were tested for normal distribution using the Shapiro-Wilk test. I would have thought that performing this test on such a small sample size is futile?

As noted in the above reply and in the Discussion section of our manuscript, the experiments shown in Figures 4b, 4c, 5a and 5b were informed by previous observations reported in the literature. Therefore, we had clear hypotheses with respect to the expected effect and the direction of that effect. For this reason, we deemed it appropriate to use one-tailed as opposed to two-tailed tests.

We agree with the reviewer that the application of the Shapiro-Wilk test for the data generated in the adhesion assays to test for normality may be futile due to the still limited number of biological replicates used in these technically challenging *in vitro* experiments. In the revised version of the manuscript, we have removed the reference to the Shapiro-Wilk test accordingly.

3. Response to Reviewer 2 Comment 5: “We agree with the reviewer that the SomaScan and immunoassay findings for APC provided in the manuscript appear to be discrepant. A likely explanation for this observation is the difference in statistical power for the analyses of the data generated with the SomaScan and immunoassays, performed in 3,301 and 52 individuals, respectively.”

Directionality should not be different because of lack of statistical power due to differences in sample sizes. The Somascan analysis might have more samples but if the measurements cannot be validated with an immunoassay, then the aptamer results are questionable and may not reflect APC levels.

We thank the reviewer for the clarification. In Figure 2c, our data showed that the confidence interval of the genotypic effect on APC levels measured using an immunoassay in our recall-study overlaps zero. Therefore, it is not possible to make claims about the directionality observed with any certainty. In particular, as we have demonstrated by a post-hoc power calculation, we likely had insufficient statistical power to detect an effect with a sample size of N=52 in the recall-study.

Our original comment on the “discrepant findings” between the SomaScan and immunoassays related to the detected differences in the statistical association rather than effect directionality.

Furthermore, as outlined in our manuscript, we performed additional experiments to confirm that the APC SOMAmer is indeed specific for APC (Table S2). However, we agree with the reviewer that further replication in another well-powered study is required to replicate our findings.

4. The response to the platelet contamination issue is also not very convincing, and the authors should discuss this in the limitation section.

We thank the reviewer for the comment. We have now added this point as a potential limitation to our revised Discussion section.

Reviewer #3 (Remarks to the Author):

The authors have addressed all of my concerns. The manuscript is much improved.

We thank the reviewer for the positive feedback on our manuscript.

Reviewers' Comments:

Reviewer #2:

Remarks to the Author:

Thank you for your response and increasing the n-numbers in key experiments. It strengthens the findings of the paper.

Elucidating mechanisms of genetic cross-disease associations: an integrative approach implicates protein C as a causal pathway in arterial and venous diseases

Rebuttal letter #3

Reviewer #2 (Remarks to the Author):

Thank you for your response and increasing the n-numbers in key experiments. It strengthens the findings of the paper.

We thank the reviewer for the positive feedback on our manuscript.